# Ultra-photostable small-molecule dyes facilitate near-infrared biophotonics

Kui Yan[1,4], Zhubin Hu[2,4], Peng Yu[1], Zuyang He[1], Ying Chen[1], Jiajian Chen[3], Haitao Sun [2] ✉, Shangfeng Wang [1] ✉ & Fan Zhang [1] ✉

Long-wavelength, near-infrared small-molecule dyes are attractive in biophotonics. Conventionally, they rely on expanded aromatic structures for redshift, which comes at the cost of application performance such as photostability, cell permeability, and functionality. Here, we report a ground-state antiaromatic strategy and showcase the concise synthesis of 14 cationic aminofluorene dyes with mini structures (molecular weights: 299–504 Da) and distinct spectra covering 700–1600 nm. Aminofluorene dyes are cell-permeable and achieve rapid renal clearance via a simple 44 Da carboxylation. This accelerates optical diagnostics of renal injury by 50 min compared to existing macromolecular approaches. We develop a compact molecular sensing platform for in vivo intracellular sensing, and demonstrate the versatile applications of these dyes in multispectral fluorescence and optoacoustic imaging. We find that aromaticity reversal upon electronic excitation, as indicated by magnetic descriptors, not only reduces the energy bandgap but also induces strong vibronic coupling, resulting in ultrafast excited-state dynamics and unparalleled photostability. These results support the argument for ground-state anti-aromaticity as a useful design rule of dye development, enabling performances essential for modern biophotonics.

Molecular dyes are important light transducers in biophotonics. Typical applications include optical contrast agents and sensors, phototherapeutics, photocatalyst, and photoswitches[1–6], while one of the universal interests among those all is achieving tunable near-infrared spectra within a small structure. This is particularly true when diminished photon scattering at near-infrared spectrum (NIR: 700–1700 nm) substantially increases tissue penetration[7–9], and a small molecular structure enables high synthetic and functionalized accessibility[10–12], as well as efficient cell permeability[13,14]. Combining these advantages, small-structure NIR dyes are promising for in vivo staining, labeling biomolecules, and sensing targets where one may demand minimal perturbation of the system[15,16], and are appealing for organ/tissue-specific diagnosis and therapy that require easy-to-alter pharmacokinetics[17–19]. However, such needs remain unmet, which really impose a fundamentally interesting challenge on dye chemistry.

The challenge centers on the lack of an atom-economic approach for narrowing down molecular band gap between the highest occupied molecular orbital (HOMO) and lowest unoccupied molecular orbital (LUMO). Common dye skeletons mostly display $[4n+2]\pi$-electron aromatic character (Fig. 1a left). In the ground state, Hückel's rule rationalizes the enhanced stability of aromatic systems[20], and thus decrease HOMO energy. This renders π-extension as a requirement to raise HOMO level for red-shifting aromatic dyes. Typical approaches including annelation[21–23], heteroatom doping[24–27], rigidification[28,29], are frequently used for the development of dyes with spectra extended beyond 1000 nm, which are in high demand for deep-tissue optical

[1]Department of Chemistry, State Key Laboratory of Molecular Engineering of Polymers, Shanghai Key Laboratory of Molecular Catalysis and Innovative Materials and iChem, Fudan University, Shanghai, PR China. [2]State Key Laboratory of Precision Spectroscopy, School of Physics and Electronic Science, East China Normal University, Shanghai, PR China. [3]Department of Breast Surgery, Key Laboratory of Breast Cancer in Shanghai, Fudan University Shanghai Cancer Center, Shanghai, PR China. [4]These authors contributed equally: Kui Yan, Zhubin Hu. ✉e-mail: htsun@phy.ecnu.edu.cn; sfwang@fudan.edu.cn; zhang_fan@fudan.edu.cn

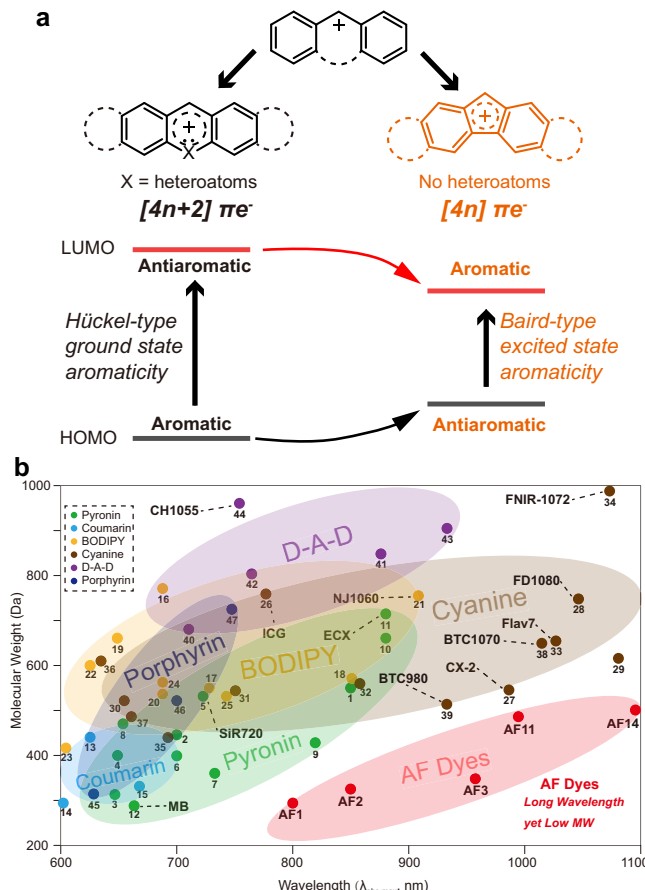

**Fig. 1 | Concept of antiaromatic dyes, and comparison between AF dyes and conventional dyes. a** The concept description of heteroatom-free antiaromatic molecular dyes. Left: common dyes with ground-state aromaticity, right: AF dyes with ground-state antiaromaticity. **b** Graphic showing the relationship between molecular weight (MW) and wavelength of different types of fluorescent dyes. All dyes were numbered, and their corresponding chemical structures and properties were shown in Supplementary Fig. 2 and Supplementary Table 1. Characters such as FD1080 and BTC1070 were adopted from their names in previous literatures.

applications. However, those approaches make dye skeletons larger (Fig. 1b, Supplementary Fig. 2), likely resulting in poor solubility, low photostability, and limited functionalization capabilities.

The limitation of aromatic system motivates us to consider its opposite, i.e., ground-state antiaromaticity. Baird in 1972 predicted that the Hückel (anti)aromaticity observed for ground-state systems is reversed in the lowest triplet state[30], and it has been computationally[31–33] and experimentally[34–37] proved that this rule likewise applies to the lowest singlet state. One may expect a naturally reduced HOMO-LUMO gap in a $[4n]\pi$-electron system because ground-state antiaromaticity energetically raises HOMO level while excited-state aromaticity decreases LUMO level[38,39] (Fig. 1a right).

Here, we explore this approach and report the insertion of a $4\pi$-electron five-membered ring into push-pull electronic structures to access a series of aminofluorene (AF) dyes with molecular weights of 299–504 Da and spectra covering 700–1600 nm. Notably, AF dyes belong to a class of dye structures that demonstrate favorable wavelength-to-molecular weight ratio reported to date (Fig. 1b, Supplementary Table 1). The introduction of antiaromaticity was found to increase vibrational coupling upon electronic transition, resulting in ultrafast excited-state decay and ultrahigh molecular photostability. Furthermore, we demonstrate many advantages of AF dyes due to their small size, such as good cell permeability and ease of functionalization to access pharmacokinetics tunability and stimulus-

responsive properties, which make them powerful tools for bioimaging and sensing.

## Results

### Molecular design and synthesis
At the outset, we devised a compact chromophore unit featuring two amine groups linked to the C3 and C6 position of a cationic fluorene skeleton, which incorporates a $4\pi$-electron five-membered ring as an antiaromatic core[40] (Supplementary Fig. 1a). This aminofluorene (AF) chromophore structurally resembles pyronin dye that has a six-membered aromatic core, but differs remarkably in frontier molecular orbital configurations. Of particular note is the formation of an anti-bonding node in the HOMO and a bonding node in the LUMO between the benzene rings (Supplementary Fig. 1b). This leads to the destabilization of the HOMO and the stabilization of the LUMO, consequently resulting in a significantly narrower HOMO-LUMO gap. To our delight, a pioneer work in 1954 by Barker et al. confirmed this assumption by reporting a 9-phenyl-substituted aminofluorene structure with near-infrared absorption beyond 900 nm[41]. However, the dye rapidly faded in protic solvents due to the electron-deficiency of ground state, and during the preparation of this manuscript, Grzybowski et al. tried a steric aryl strategy to stabilize the cationic fluorene skeleton in water[42]. Despite these preliminary explorations, whether the concept of anti-aromatic dyes can be implemented in reductive biological environment and generates biological and photonic benefits remains uncertain.

Here, we synthesized the sterically-protected and substituent-diversity AF dyes (Fig. 2a, b) in a concise two-step route, which began from the Buchwald-Hartwig C-N cross-coupling of nitrogen nucleophiles with the same substrate dibromofluorenone, followed by aryl-lithium addition to install sterically hindered aryl motif. This route can efficiently access 14 dyes (AF3-AF14) with different substituents, giving gram-scale products in 36–79% overall yields. The cross-coupling was tolerant of carbamates, enabling the synthesis of AF1 (299 Da) of the lowest molecular weight and AF2 through a post-deprotection approach (Supplementary Fig. 3a). All structures were identified by $^1H$ and $^{13}C$ NMR spectroscopies and mass spectrometry (Supplementary Figs. 33–112).

### Structure-property relationship
The photophysical properties of AF dyes were studied in dichloromethane (DCM) (Fig. 2b–f, Table 1). We plotted the absorption maxima and corresponding substituent patterns on a spectrum axis for intuitive comparison (Fig. 2b). AF dyes spectrally cover 700–1600 nm, with their absorption maxima ($\lambda_{abs}$) spanning from 800- to 1101-nm, and emission maxima ($\lambda_{em}$) from 933- to 1187-nm, respectively (Fig. 2c, d and Supplementary Fig. 4). The '0-1' side bands of the absorption spectra are apparently large in AF1 and AF2, and reduced upon absorption redshift, which indicate varying degree of vibrational coupling upon electronic excitation. This trend accompanies with increased oscillator strength, for instance, the increase in molar absorptivities ($\varepsilon$) follow the order of AF1 (~$9 \times 10^3 M^{-1}$ cm$^{-1}$) < AF2 (~$1.5 \times 10^4 M^{-1}$ cm$^{-1}$) < AF3 (~$2.3 \times 10^4 M^{-1}$ cm$^{-1}$) < AF14 (~$4 \times 10^4 M^{-1}$ cm$^{-1}$) (Table 1). Theoretical calculations of $S_0$-$S_1$ excitation agree well with this result, revealing that electron delocalization of HOMOs and LUMOs on amine motifs promotes absorption redshift and enhancement (Supplementary Fig. 5). We further plotted $\lambda_{abs}$ as well as the peak-shoulder ratio (the ratio of 0-0 and 0-1 absorption) of absorption band against the vertical ionization energies (VIEs) of amine motifs. Excellent linear correlation was observed, apart from the primary and secondary amine of aliphatic motifs (AF1, AF2), and three aromatic amines (AF12, AF13, AF14) (Fig. 2e, f). This linear deviation can be attributed to the spectral broadening induced by the vibronic coupling of the AF skeleton, and this effect decreases as the electronic delocalization of amine motifs increase (Supplementary Figs. 5 and 6). In addition, substituents in pendant aryl group finely tune

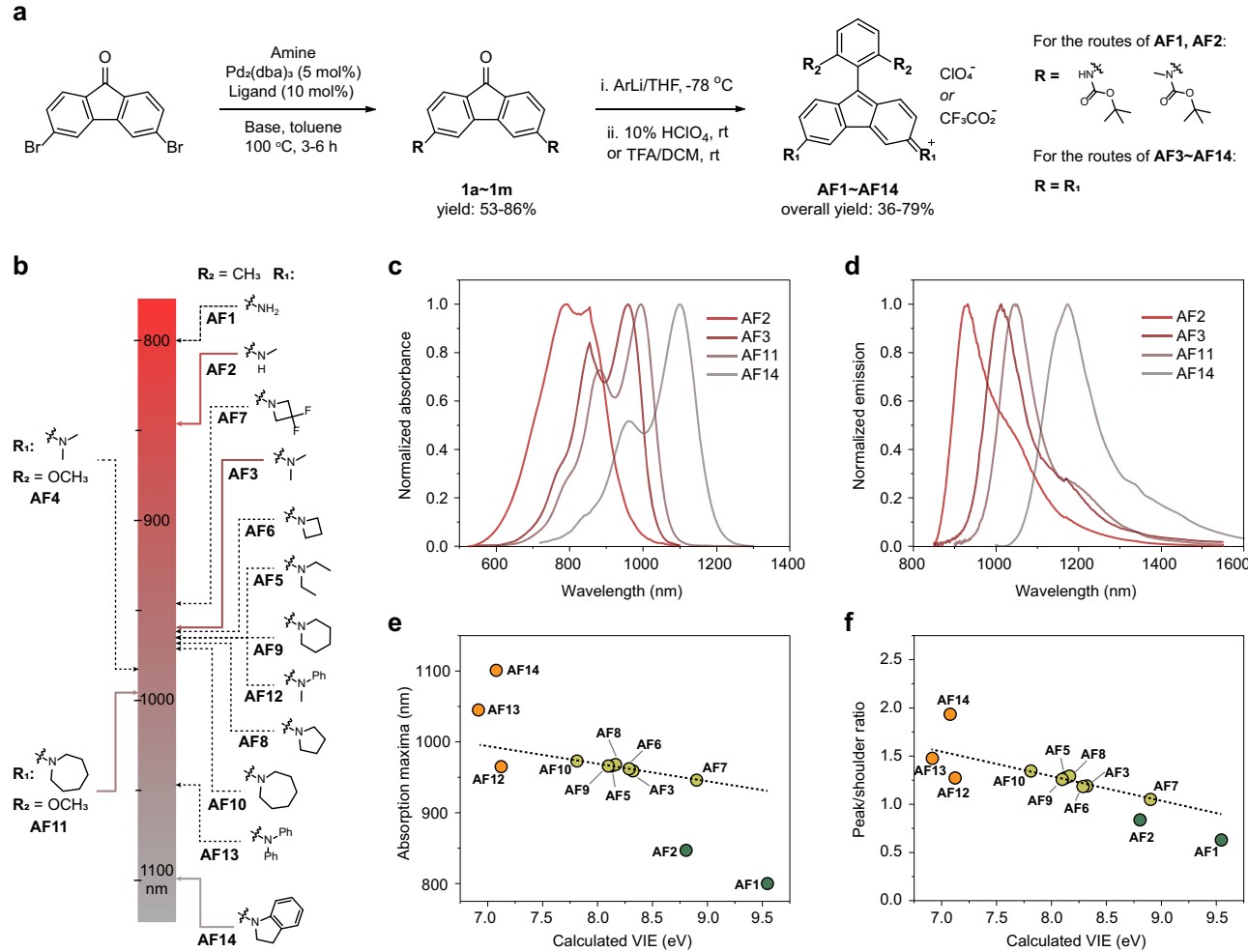

**Fig. 2 | Synthesis, spectral characteristics and structure-property relationships of AF dyes. a, b** Two-step synthetic route for AF1–AF14 **a** and their corresponding substituent patterns **b** with graphic showing the absorption maxima on a spectrum axis. Representative spectrally-distinctive dyes (AF2, AF3, AF11 and AF14) were shown in colored solid lines. **c, d** Normalized absorption **c** and emission **d** spectra of AF2, AF3, AF11 and AF14. **e, f** Plots indicating the linear relationship (AF1, AF2, AF12, AF13, and AF14 were excluded from linear fitting) between absorption maxima **e** peak-shoulder ratio **f** of AF dyes and calculated vertical ionization energies of amine substituents. All spectral properties were collected in dichloromethane.

the absorption maxima (AF3: methyl-, $\lambda_{abs}$ = 959 nm; AF4: methoxy-, $\lambda_{abs}$ = 983 nm), suggesting weak electronic coupling between aryl group and fluorenone skeleton. Overall, the structure-absorption relationship resembles that observed from xanthene-type dyes[43], implying a similar rule for the substituents to tune the absorption of antiaromatic and aromatic dyes. Notably, all AF dyes are stable in aqueous solution (phosphate buffer, pH 7.4) with constant absorption spectra (Supplementary Fig. 4), indicating the steric bulk structure is effective to protect electrophilic AF skeleton.

Trends in structure-emission relationship of AF dyes substantially differs from that explored in xanthene-type dyes with same substituent replacement (Table 1, Supplementary Fig. 4). For example, the smallest AF1 has a largest Stokes shift of 176 nm and lowest quantum yield (Φ) of 0.003%. AF2 with methylamine substituents has a highest Φ of 0.023% and a moderate Stokes shift of 86 nm. Azetidine (AF5, 0.013%) and indoline (AF14, 0.008%) substituents result in similar Φ as dimethylamine substituents (AF3, 0.011%). In addition, the Φ of AF3 also shows no significant difference in different solvents (Supplementary Table 2). However, xanthene-type dyes with same substituent patterns display small Stokes shifts, large changes in Φ, and solvent-dependent properties owing to the twisted intramolecular charge

transfer (TICT) of amine motifs[44,45]. These results suggest a distinct decay route for the excited states of AF dyes, not associated with TICT.

## Mechanistic insight into the photophysics

To gain insights into the nature of photophysics, AF3 was selected and investigated as an example. The excited-state dynamics of AF3 was studied using femtosecond transient absorption (fs-TA) spectroscopy (Fig. 3a). Using 810 nm pump, an unambiguous negative band appeared at approximately 825–1200 nm, and rapidly decayed to the ground state within 50 ps. Comparison with the steady-state absorption and emission spectra indicates that the near-infrared negative band can be assigned to the superposition of the ground-state bleaching (GB) signal and stimulated emission (SE) band of the first excited state to ground state ($S_1$-$S_0$) transition. We did not observe near-infrared excited-state absorption (ESA) signal, but using 375 nm pump, ESA signal appeared at 500-700 nm within the identical time scale (Supplementary Fig. 7), coinciding with the first excited state to higher excited states ($S_1$-$S_n$) transition. Global analysis of the near-infrared fs-TA data gave a decay-associated difference spectrum (DADS) consisting of one species of $\tau$ = 7.1 ps. This species correlates well with the single-wavelength kinetic fit at 977 nm (Fig. 3b),

**Table 1 | Photophysical properties of AF Dyes in DCM**

| Dye | Molecular weight[a] (g/mol) | $\lambda_{abs}^b$ (nm) | $\lambda_{em}^b$ (nm) | $\varepsilon^c$ (M⁻¹ cm⁻¹) | Peak-shoulder ratio | Stokes Shift (nm/cm⁻¹) | Quantum Yield (10⁻² %) |
|---|---|---|---|---|---|---|---|
| AF1 | 299 | 800 | 976 | 9047 | 0.629 | 176/2254 | 0.3 |
| AF2 | 327 | 847 | 933 | 14869 | 0.987 | 86/1088 | 2.3 |
| AF3 | 355 | 959 | 1014 | 23052 | 1.189 | 55/565 | 1.1 |
| AF4 | 387 | 983 | 1046 | 19795 | ND[d] | 63/613 | 0.6 |
| AF5 | 411 | 966 | 1015 | 25261 | 1.274 | 49/500 | 1.3 |
| AF6 | 379 | 962 | 1014 | 21685 | 1.184 | 52/533 | 1.3 |
| AF7 | 451 | 946 | 995 | 15231 | 1.050 | 49/521 | 1.1 |
| AF8 | 407 | 968 | 1017 | 24705 | 1.291 | 49/498 | 1.3 |
| AF9 | 435 | 966 | 1026 | 26793 | 1.258 | 60/605 | 1.4 |
| AF10 | 463 | 973 | 1026 | 22143 | 1.345 | 53/531 | 1.2 |
| AF11 | 495 | 994 | 1045 | 21081 | ND[d] | 51/491 | 1.2 |
| AF12 | 479 | 965 | 1029 | 18205 | 1.273 | 64/645 | ND[d] |
| AF13 | 603 | 1045 | 1196 | 30777 | 1.476 | 151/1208 | 0.5 |
| AF14 | 503 | 1101 | 1187 | 41132 | 1.933 | 86/658 | 0.8 |

[a]Molecular weight of AF dyes excluding counter ion, [b]absorption and emission maxima, [c]Molar absorptivity, [d]ND=not determined.

indicating that the excited-state dynamics of AF3 merely involves the transition from $S_1$ to $S_0$ state. Regarding to this ultrafast relaxation, intersystem crossing and other photochemical processes are not competitive (Fig. 3c). Using the excited-state lifetime, the radiative ($k_r = \Phi/\tau$) and nonradiative ($k_{nr} = (1-\Phi)/\tau$) rate constants of AF3 were calculated. While the $k_r$ ($1.5 \times 10^7 \, s^{-1}$) is at the same level as compared with reported NIR-II dyes[46], the $k_{nr}$ ($1.4 \times 10^{11} \, s^{-1}$) increases by 1 or 2 orders of magnitude. This result indicates that rapid internal conversion is primarily responsible for the excited-state dynamics of AF dyes.

To understand the ultrafast excited-state dynamics, the geometric structures of AF3 at the $S_0$ and $S_1$ states were extensively studied. The X-ray crystal structure of AF3 reveals that the fluorene skeleton forms a rigid plane, with the dihedral angle between the mean planes of the two benzene rings to be 2.63° (Fig. 3d, Supplementary Table 3). The 2,6-dimethylphenyl group at the 9-position lies 65.4° out of the plane. The C3-C4 bond that connects the two benzene rings in the fluorene skeleton is distinctly elongated at 1.486 Å, surpassing the lengths of other C-C bonds in the fluorene structure. These observations were confirmed in the optimized $S_0$ geometry (Fig. 3e, f) based on the density functional theory (DFT) calculations. While for the optimized $S_1$ structure, nine C-C bonds undergo substantial bond length changes, centralizing on one side of the fluorene skeleton (C13-C1, C1-C2, C2-C3, C3-C4, C4-C5, C5-C6, C6-C7, C9-C4, C3-C11), with each bond experiencing a change in length between the $S_0$ and $S_1$ states exceeding 0.025 Å. We hypothesized that this bond length redistribution is linked to the changes in aromaticity, as for all AF dyes. Further, both nucleus-independent chemical shift scans along the X and Y axes (NICS-XY scan) and calculations of anisotropy of the induced current density (AICD) indicate a highly localized antiaromaticity in the five-member ring of the $S_0$ state, coupled with increased aromaticity in the entire fluorene skeleton of the $S_1$ state, although local circulations in the current density at the transannular bonds seem to slightly weaken the aromatic character in the 5-membered ring. (Fig. 3g–i, Supplementary Figs. 8–21). These results suggest the presence of an aromaticity reversal-induced vibronic coupling in the $S_1$-$S_0$ nonradiative transition. Moreover, a series of electronic indexes used for characterizing (anti-)aromaticity such as multicenter index (MCI) and electron localization function-pi (ELF-π) are calculated (Supplementary Table 4, Supplementary Fig. 22). It can be seen that the magnetic indices NICS and current densities from AICD indeed suggest a reversal in the antiaromatic and aromatic upon excitation from $S_0$ to $S_1$, while the results

from these electronic indexes (i.e. MCI and ELF-π) do not (see Supplementary Notes for more details). For the consistency and difference between the magnetic and electronic indices used for characterizing (anti-)aromaticity, we refer the interested readers to, e.g., refs. 39,47,48.

Based on the thermal vibration correlation function formalism coupled with first-principles DFT calculations[49], the internal conversion rate can be expressed in two parts, namely the nuclear motion term defining the contribution from nuclei and the electronic coupling term defining the contribution from electrons. We then calculated the reorganization energy and electronic coupling factor of AF3, as a function of each normal mode. Figure 3j, k demonstrate that the stretching vibrational modes of C-C bonds at 1000–1700 cm⁻¹ strongly contribute to the large reorganization energy and electronic coupling. This highlights the significant correlation between bond length redistribution and electronic/vibrational degrees of freedom. We further analyzed the weighting factor for each vibration contributing to the nonradiative decay rate. Figure 3l shows the internal conversion rate ($k_{ic}$) of each promoting mode, and a calculated total $k_{ic}$ ($1.06 \times 10^{11} \, s^{-1}$) agrees well with the experimental value. It was observed that the total contribution of C-C stretching mode to $k_{ic}$ is much greater than that of the high-frequency C-H stretching mode. Note that the latter has been quantified as an important factor limiting the emission performance of polyatomic molecules beyond 1000 nm in the context of the energy gap law[46]. Hence, the results collectively reinforce the argument that the aromaticity reversal of the fluorene skeleton establishes C-C vibrational coupling as the primary driver of the nonradiative transition, which accounts for the ultrafast excited state dynamics and low quantum yields of AF dyes.

## Investigation of cell permeability and photostability
With the AF dyes in hand, we further explored the advantages of their small size in biological applications. It is necessary to examine their chemical stability toward biological environment, in particular, in the presence of glutathione (GSH), the most abundant nucleophilic thiols in living cells. We plotted the dose-response curves of all AF dyes as a function of GSH concentration (Supplementary Fig. 23). All dyes survived under the intracellular GSH condition (1–10 mM), though, the absorbance of AF4, AF7, and AF13 showed a decline when the GSH concentration exceeded 100 μM. We attributed this phenomenon to the reduction in the electron-donating ability of difluoroazetidine and diphenylamine, as well as the decreased steric hindrance of the 2,6-dimethoxy aryl group compared to the 2, 6-dimethyl aryl group. After identifying the low cytotoxicity of AF2 and AF3 within dye concentration of 10 μM through CCK-8 assay (Supplementary Fig. 24), we unexpectedly found that AF dyes brightly stained cells at the NIR-II spectrum (1100 nm long-pass filter) in as little as 30 s of co-incubation (Fig. 4a, Supplementary Fig. 25). Despite the limitations of immature NIR-II colocalization techniques, we speculated that these signals primarily occured in mitochondria, based on their punctate distribution surrounding the cell nucleus. In any case, this character makes it practical to study cell-dye interaction property through near-infrared fluorescence imaging. To illustrate the correlation between cell staining and molecular size, a commercially available NIR-II dye IR1048[50], with a molecular weight of 653 Da, was employed for cell imaging comparison under the same excitation and fluorescent acquisition condition as AF3. AF3 efficiently stained cells with a high signal-to-background ratio (SBR) of approximately 80 within just 10 min, whereas IR1048, even after 30 min of co-incubation, failed to enter the cells, resulting in no observable signals (Fig. 4b, c). Quantitative analysis of the extracellular fluid reveals that >90% of AF3 permeates into cells, which is ~42-fold higher than that of IR1048. The results illustrates the advantages of small structure in cell permeability and staining intensity.

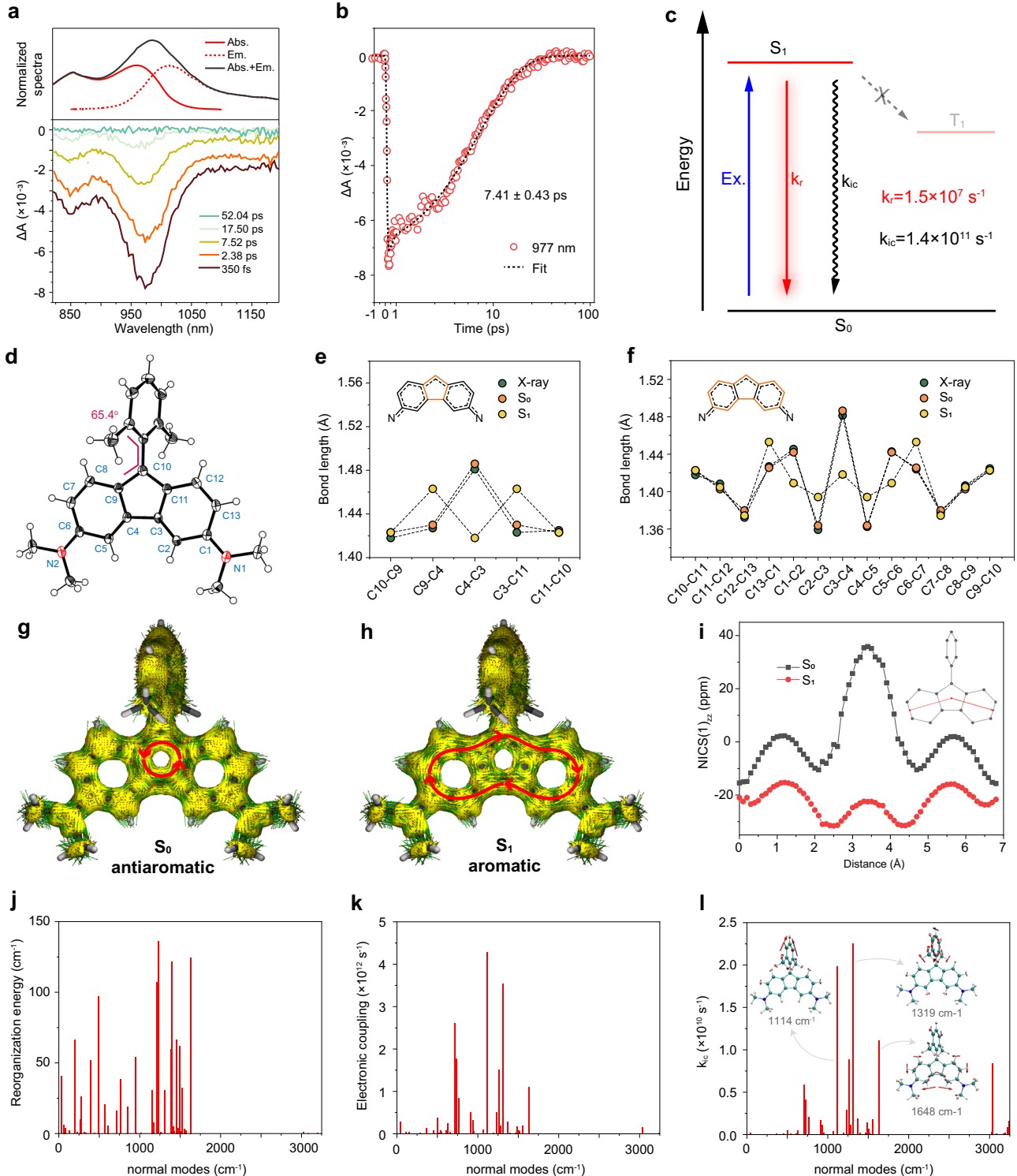

**Fig. 3 | Photophysical mechanistic investigation of AF3. a, b** fs-TA spectrum of AF3 **a** and **b** single-wavelength kinetic fit result at 977 nm. Solvent: dichloromethane. **c** Jablonski diagram of AF3 excitation process. **d** X-ray crystal structure of AF3. **e, f** AF3 bond length comparison of X-ray structure and $S_0/S_1$ optimized structures on the central 5-membered ring and the whole fluorene skeleton shows strong BLA. **g, h** Theoretical calculated AICD diagram of AF3 in $S_0$ **g** and $S_1$ **h** state. For high-resolution AICD plots of all AF dyes, please refer to Figs. S7-S20 in the Supplementary Information. **i** NICS(1)$_{zz}$-XY scan result of AF3 in both $S_0$ and $S_1$ state. The red line in the inset shows the scanning path with a scanning interval of 0.1 Å. **j–l** Calculated reorganization energy **j**, **k** electronic coupling factor, and **l** $k_{ic}$ versus normal vibration mode (wave numbers) for $S_1$ state of AF3. Representative normal modes were shown as insets.

Photobleaching presents a major obstacle when revisiting staining samples. Therefore, we conducted in vitro light exposure assay to examine the photostability of AF dyes. In aqueous solution, AF dyes (AF2, AF3, AF11, AF14) showed no changes in absorbance or chemical structure after 30 min of laser exposure (808 nm for AF2, AF3, AF11,

and 980 nm for AF14 with power density of 1.6 W cm$^{-2}$), while Cy7, a commonly used NIR dye experienced rapid absorbance attenuation within 3 min of exposure to 808 nm laser at the same power density (Supplementary Fig. 26). However, with the same light exposure, we did not observe photobleaching of IR1048 in organic solvents (IR1048

is not water-soluble), which may be attributed to the low band gap of NIR-II dye minimizes the production of photooxidative species. For a thorough comparison, AF3 and IR1048 were subjected to a microscope assay in which the dyes were deposited onto a slide, exposed to air, and received intense laser irradiation (808 nm for AF3 and 980 nm for IR1048 with power density of 200 kW cm$^{-2}$) and simultaneously recorded the fluorescence changes. It was observed that after 2 h of irradiation, AF3 retained nearly 90% of its fluorescence signals, whereas IR1048 exhibited a decay exceeding 50% (Supplementary Fig. 26). The exceptional photostability of AF3 was further demonstrated in continuous cell imaging, allowing for over 1000 repetitions of snapshot recording with intense 808 nm laser irradiation (200 kW cm$^{-2}$). Notably, the images kept relatively high and stable SBR (Fig. 4d, Supplementary Movie 1). In sharp contrast, when Cy7-stained cells were repetitively excited with 808 nm at the same power density, they underwent complete bleaching in fewer than 15 frames (Fig. 4e, Supplementary Movie 2). Overall, these results highlight the ultraphotostability of AF dyes, which hold significant promise for optical imaging in challenging conditions.

## Altering pharmacokinetics and biodistribution of AF dyes for in vivo fluorescence bioimaging

The small structures of AF dyes offer great flexibility for chemical modifications. This is highly convenient for developing contrast agents with finely tuned pharmacokinetics and biodistribution, which may find extensive applications in organ/tissue-specific theranostics and clinical translation. To demonstrate the feasibility, we made a minor alteration—a net addition of one carboxyl group—to AF2 and AF3 (Fig. 5a), respectively, which brings a small increase (+44 Da) in the molecular weights, but elicits a substantial decrease (-4.19) in the oil-water partition coefficients (cLog D) at pH 7.4 (Supplementary Table 5, Supplementary Fig. 3b). Calculation found that more than 99% of the resulting AF2-COOH and AF3-COOH exist in the zwitterionic forms at pH 7.4, implying a reduced unspecific interaction with biological

species. This is impressive as the majority of existing NIR dyes requires complex macromolecular modifications[51,52] or micelle formulations to obtain water solubility with drawbacks such as tedious preparation, undesired pharmacokinetics or serum instability[53]. For a direct comparison, we synthesized an AF3-Dextran conjugate with a molecular weight of approximately 43300 Da (Supplementary Fig. 3c), which represents a conventional water-soluble modification approach for large-structure dyes.

We then subjected AF3, AF3-COOH and AF3-Dextran to in vivo fluorescence imaging to track their biodistribution over time (Fig. 5b–e). At first 20 min post intravenous (i.v.) injection, AF3 rapidly outlined liver and intestinal tract, which may stem from its efficient cell permeability. AF3-COOH briefly appeared in liver for one minute, and rapidly metabolized to small intestine with accumulative signals in bladder, indicating a composite metabolic pathway including hepatoenteral circulation and renal clearance. AF3-Dextran had a similar fluorescence pattern with AF3-COOH but showed observable signals in blood vessels and long-lasting signals in liver, which is reasonable because dextran has a broad molecular weight distribution. In the following 80 to 100 min, liver signals of all three compounds were metabolized, and the signals of AF3-COOH and AF3-Dextran in the bladders reached a bright plateau until urination occurred. The results underscore the considerable impact of molecular structure and molecular weight variations on the pharmacokinetics and biodistribution of AF dyes.

Inspired by its rapid renal clearance feature, we were driven to investigate the potential applications of AF3-COOH in pre-clinical diagnosis. For example, renal ischemia-reperfusion (RIR) is a critical clinically acute renal injury with high mortality, and currently there is a lack of reasonable surveillance techniques for the rapid and early detection of RIR[54,55]. Previous studies have employed the clearance hindrance of water-soluble fluorescent macromolecules in injury kidney for optical detection of RIR, but it is time-consuming to reach a high SBR for accurate diagnosis due to their slow renal clearance[56]. With the anticipation that AF3-COOH could potentially rapidly

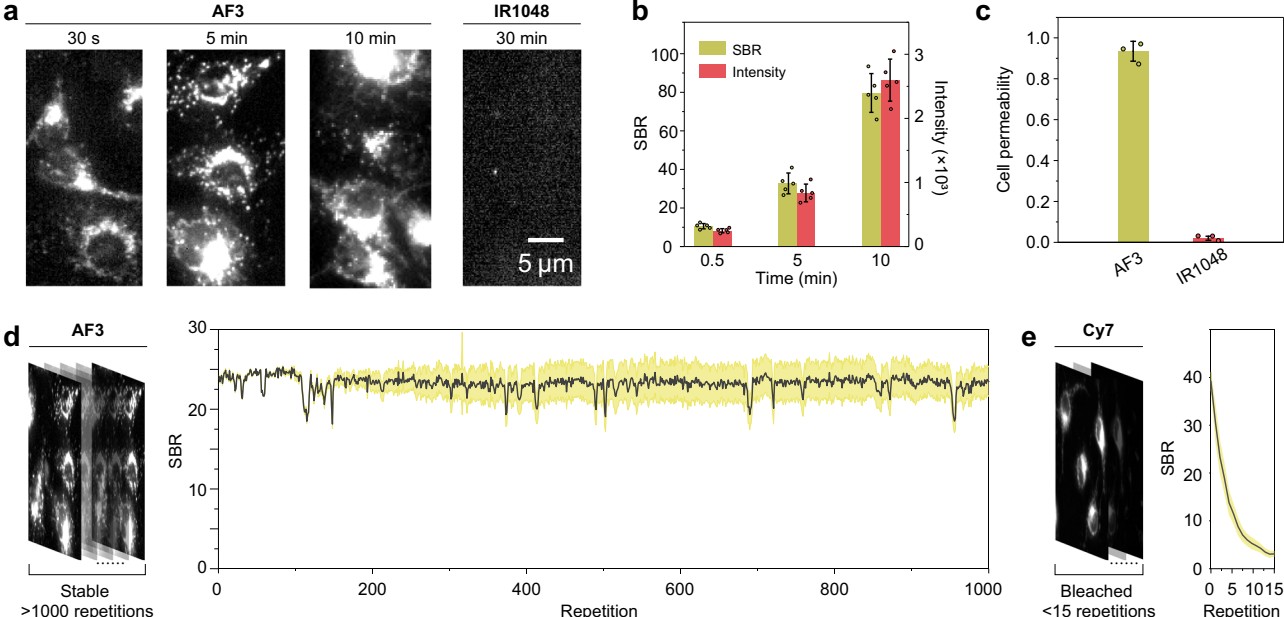

**Fig. 4 | Cell permeability and photostability characterization. a** Fluorescence images of dye-stained endothelial cells (ECs). AF3: staining for 30 s, 5 min, and 10 min. IR1048: staining for 30 min. Scale bar, 5 μm. **b** Quantified data of the signal-to-noise ratio (SBR) and mean fluorescence intensities for AF3-stained cells at different time points. The bars represent mean ± s.d. derived from $n = 5$ independent cells. **c** Quantified data of the cell permeability for AF3 and IR1048. The bars represent mean ± sd derived from $n = 3$ independent experimental groups.

**d** Repetitive fluorescence recording of AF3-stained endothelial cells at a laser power density of 200 kW cm$^{-2}$. Right panel shows the signal-to-noise ratio (SBR) of cell images versus irradiation time. Dips in SBR curve were mainly caused by the changes of the instrument focal plane. The black lines and yellow bars represent mean ± sd derived from $n = 5$ independent cells. **e** Control result using Cy7-stained cells. The black lines and yellow bars represent mean ± sd derived from $n = 5$ independent cells.

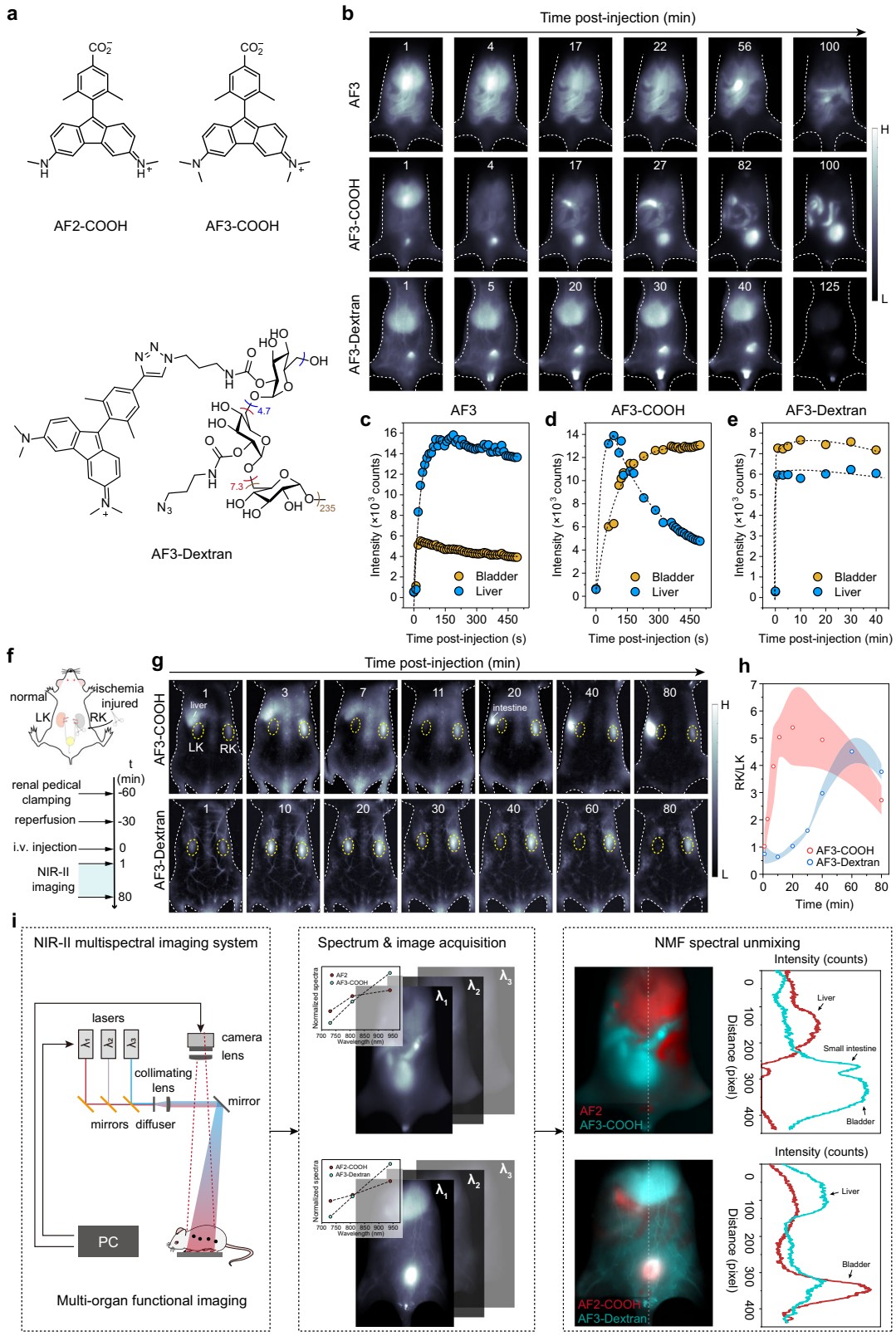

diagnose RIR, we intravenously injected AF3-COOH in a mouse model (Fig. 5f–h), featuring RIR in the right kidney (RK) and a healthy left kidney (LK). The same procedure was applied to AF3-Dextran, serving as a macromolecular control. It was observed that the fluorescent signals of AF3-COOH were rapidly cleared from the healthy kidney in 11 min, whereas this process took 60 min for AF3-Dextran (Fig. 5g, Supplementary Fig. 27). Conversely, both agents gradually accumulated in the RIR kidney, resulting in enhanced fluorescence signals during the 60 min following injection. Therefore, AF3-COOH achieved a SBR (defined as the RK/LK ratio) surpassing the Rose criterion (SBR > 5) for 100% certainty[7] within 11 min, which outperformed AF3-Dextran by sixfold in speed (Fig. 5h). These results demonstrate the great value of AF dyes in clinical diagnosis, with the potential to advance diagnosis time and increase biocompatibility.

**Fig. 5 | Functionalized AF dyes for in-vivo fluorescent imaging. a** Chemical structures of AF2-COOH, AF3-COOH and AF3-Dextran. **b**, In vivo NIR-II fluorescence images of mice at different time points after i.v. injection of AF3, AF3-COOH and AF3-Dextran, respectively. **c–e** Quantified data of the mean fluorescence intensities of bladders and livers for the images in **b**. **f** Schematic illustration of the ischemia-reperfusion process and in vivo NIR-II imaging. Right kidney was 30 min ischemia treated. **g**, In vivo NIR-II fluorescent images of renal ischemia-reperfusion (RIR) mice from the dorsal position at different time points after intravenous injection of each probe. The dashed circles indicate the kidney position. The bright fluorescence signals near the left kidney of AF3-COOH-treated group were mainly from liver (1 to 7 min) and intestine (20 to 80 min). **h** Time-dependent evolution of the

fluorescence intensity ratio of right-to-left kidney (RK/LK) for the images in **g**. The data are the mean ± s.d. derived from $n = 3$ biologically independent mice. **i** Work flow of NIR-II multispectral fluorescence imaging. Multiple lasers were coupled into a wide-field NIR-II fluorescence imaging system, and excitation and emission collection were implemented under the control of a digital delay/pulse generator. Inset plots in the middle panel show the excitation spectra of AF2 and AF3 acquired from their administered solutions, which were then used for the NMF spectral unmixing of the image stacks generated from in vivo data. Right panel shows the RGB false color representation of unmixing results, and inset plots are the cross-sectional intensity profile along the center lines of images, which show the fluorescence distribution in multiple organs.

The flexibility of AF dyes in spectrum and pharmacokinetic tuning enables us to showcase the utility of AF dyes in multiplexed imaging. A home-built wide-field NIR-II multispectral imaging implementation was established for technical demonstration (Fig. 5i left), in which three wavelengths of lasers (here we used 730, 808, and 940 nm) were used to excite a sample in a stepwise manner and fluorescence was collected at one NIR-II channel (>1100 nm). Prior to animal imaging experiment, we recorded the excitation spectra of AF2 and AF3 (Fig. 5i middle), and used the spectra to unmix the image stacks with nonnegative matrix factorization (NMF) decomposition algorithm. After we identified a minimum signal crosstalk (less than 5%) in each channel (AF2/AF3) (Supplementary Fig. 28), we employed i.v. injection to administer a mixture of AF2 and AF3-COOH or a mixture of AF2-COOH and AF3-Dextran to two groups of mice, respectively, and further performed multispectral imaging at 30 min post injection. Unlike the mixed message in single spectral channel, NMF spectral unmixing clearly presented the multiple metabolic pathways in a same mouse, which was stemming from the pharmacokinetic-diversified characters of contrast agents (Fig. 5i right). Collectively, these experiments verifies the advantages of AF dyes on structure modification, demonstrating potential for in vivo imaging.

## Mini sensing platform based on AF skeleton

Small-molecule sensors are more acceptable by developers in view of their easy synthetic handling and modification, good solubility and biocompatibility. We thus envision a mini sensing platform based on AF dyes, which would be a pioneering example of NIR optical probes based on antiaromatic system.

We found AF2 was readily deprotonated in mild alkaline environment, resulting in a blue shift of absorption peak from 875 to 590 nm (Supplementary Fig. 29a, b). As such, the acetylation of AF2 can smoothly afford the model compound AF2Ac (Supplementary Fig. 29c), which simulates the activatable probe based on amide breaking, a general mechanism used for the sensing of reactive oxygen species (ROSs) and proteases. Absorbance of AF2Ac around 500–1000 nm disappears at pH 7 (Supplementary Fig. 29b) as compared with that of AF2 and the deprotonated form (AF2 at pH 13), suggesting the loss of large delocalized π-structure. The spectra of AF2Ac and AF2 are fairly stable at a pH range of 1–9, implying AF2 is promising for reliable biosensing (Supplementary Fig. 29d). As a proof of concept, we synthesized AF2B (Fig. 6a) with a borate motif sensitive to peroxynitrite (ONOO⁻), which is a biomarker overproduced in inflamed tissues. Upon the addition of various concentrations (0–0.5 equivalent) of ONOO⁻, absorbance of AF2B in PBS 7.4 gradually evaluated around 700–1000 nm and reached a plateau in seconds (Fig. 6b). The corresponding emission at 808-nm excitation increased by 20-fold as the concentration of ONOO⁻ reached to 0.5 equivalent (eq.) (Supplementary Fig. 30), and displayed high selectivity over other reactive species, including ClO⁻ (50 eq.), $H_2O_2$ (50 eq.), •OH (20 eq.), and •$O_2^-$ (50 eq.) (Supplementary Fig. 31). All these results are indicative of a qualified response to biologically relevant ONOO⁻.

The spectral changes of AF2B further encouraged us to demonstrate its broad utilities in other optical imaging mode, for example, as an acoustogenic platform for multispectral optoacoustic tomography (MSOT). First, we tested the photoacoustic (PA) readouts of several AF dyes (AF1, AF2, and AF3 were tested due to the limit of instrumental excitation) with a water-soluble PA benchmark indocyanine (ICG, commercially available). In a parallel comparison, AF dyes exhibited stable PA spectra shape correlating with molecular absorption and showed linearly increasing PA signals in a wide range of dye concentrations (50–200 μM), while ICG displayed distorted PA spectra shape and nonlinear signals (Supplementary Fig. 32). This result suggests that the good solubility of AF dyes in water confers stable spectral behavior, which are suitable for in vivo MSOT. Following this experiment, the reaction of AF2B with ONOO⁻ produced an OFF-ON-type PA signal along the excitation spectrum of 680–970 nm (Fig. 6c), showing the potential for the photoacoustic detection of deep-tissue ONOO⁻.

Next, we examined the performance of AF2B in cellular and animal imaging. Upon incubation of brain capillary endothelial cells (BCECs) with AF2B, we were able to observe a very weak staining pattern via fluorescence microscopy, which suggests our probe could not be activated by normal ONOO⁻ level in cells. After ONOO⁻ treatment for 0.5 h, 9-fold higher bright fluorescence was observed. To confirm these results, we added uric acid, an ONOO⁻ scavenger to a group of ONOO⁻-pretreated BCECs and stained the cells with AF2B. Relative to ONOO⁻-treated group, the signal intensity was 3-fold lower, confirming the activation of AF2B in cells (Fig. 6d). Next, we established a repeatable traumatic brain injury (TBI) model in the right hemisphere of mice, as ONOO⁻ overproduced in hours post TBI. AF2B was intravenously injected to the TBI mice at 12-h post injury, followed by MSOT at half an hour post administration (Fig. 6e, f). A remarkable difference in total PA signals was observed in the left (normal) and right (TBI) hemisphere of a same mouse. Through a spectral unmixing approach, the signal of AF2 can be isolated from other major light-absorbing substances, e.g., deoxyhemoglobin ($Hb_R$) and oxygenated hemoglobin ($HbO_2$). We observed an 8-fold increase in the $Hb_R$ signal and a 2.4-fold decrease in the $HbO_2$ signal of the TBI region compared to the normal region, accompanied with a 6-fold increase in the AF2 signal, which pointed to the sequentially occurred events in TBI, such as the damage of blood-brain barrier, blood coagulation, hypoxia and ONOO⁻ overproduction[57]. Overall, the results demonstrate a powerful acoustogenic platform based on mini AF dyes.

## Discussion

Small-molecule NIR dyes with a high wavelength-to-molecular-weight ratio (WMR) hold promise for a wide range of biophotonic applications, as they have the advantages in solubility, photostability, functionalization capabilities, and bioavailability. In the present work, we have developed a series of AF dyes with a $[4n]\pi$-electron Hückel antiaromatic skeleton, achieving an increased WMR of 2–2.7 ($\lambda_{abs}$/Mw), surpassing that of most of conventional dyes (WMR < 2) (Supplementary Table 1). The AF skeleton exhibits ground-state antiaromaticity and

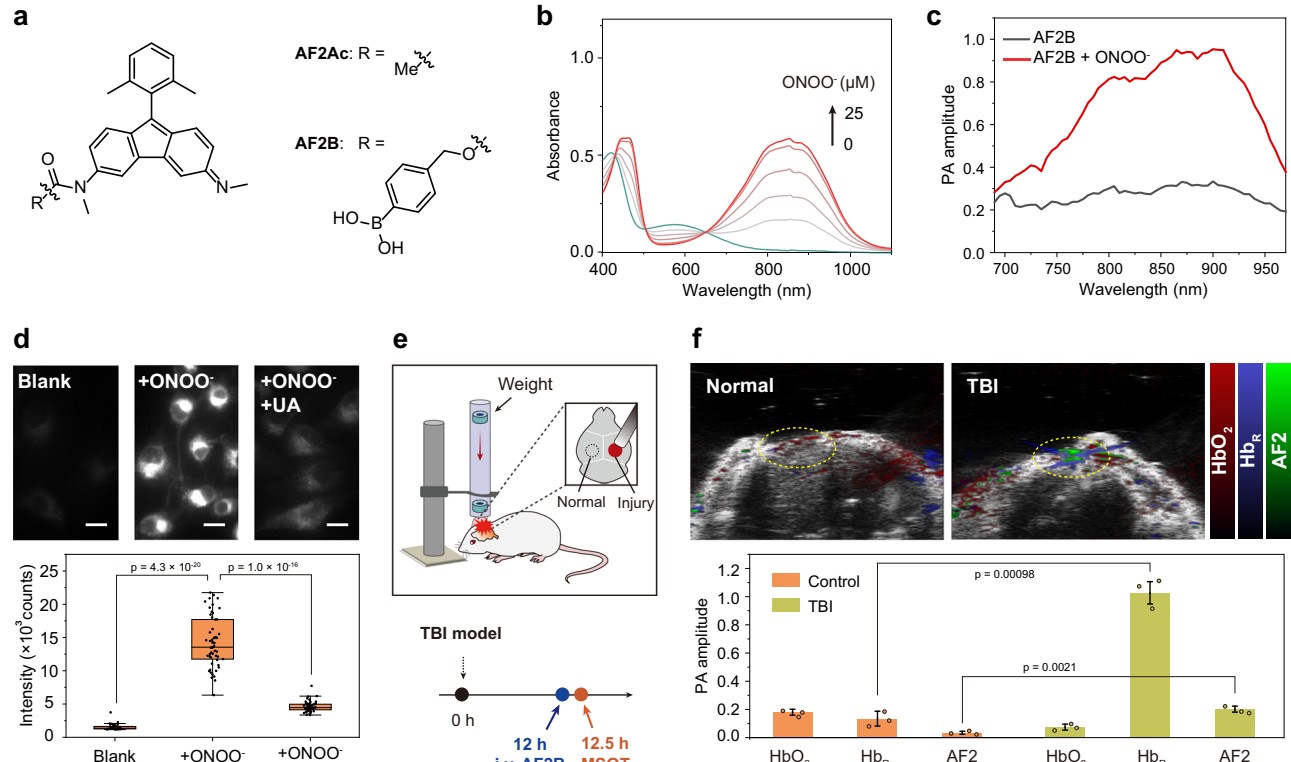

**Fig. 6 | Mini sensing platform based on AF2 for in-vivo ONOO⁻ detection.**
**a** Chemical structures of AF2Ac and AF2B. **b** Variation in absorption spectra of AF2B (50 μM) in PBS pH 7.4 after incubation with increasing concentration of ONOO⁻ (0−25 μM, gradient: 5 μM). **c** PA spectra of AF2B (50 μM) in PBS pH 7.4 before and after the addition of 25 μM ONOO⁻. **d** Fluorescence images of AF2B-stained brain capillary endothelial cells (BCECs) pre-treated with a vehicle control (Blank), or 1 μM ONOO⁻, or 1 μM ONOO⁻ + 400 μM uric acid (UA). The bars represent mean ± sd derived from $n = 35$ independent cells. The boxplot is centered at median and displays interquartile ranges (IQR) and minima and maxima extending to 1.5× IQR. Statistical analysis was performed using two-tailed Student's t-test ($\alpha = 0.05$). Scale bar, 10 μm. **e** Schematic illustration of the construction of TBI mice model and the timeline for probe administration and MSOT imaging. **f** MSOT/US images of OxyHb, DeoxyHb, and AF2 in the left (normal) and right (TBI) hemisphere of a same mouse after i.v. injection of AF2B. The bars represent mean ± s.d. derived from $n = 3$ biologically independent mice. Statistical analysis was performed using two-tailed Student's t-test ($\alpha = 0.05$).

excited-state aromaticity, which is opposite from conventional dyes with $[4n + 2]\pi$-electron Hückel aromatic character. This characteristic results in an inherently narrow HOMO-LUMO gap, with the destabilized antiaromatic $S_0$ state elevating the HOMO energy while the stabilized aromatic $S_1$ state reduces the LUMO energy. It is worth noting that the influence of ground-state antiaromaticity on wavelength differs from the $S_1$ antiaromaticity relief mechanism recently disclosed in single benzene fluorophores[12]. The latter are aromatic in their $S_0$ and antiaromatic in their $S_1$. Their $S_1$ states undergo geometry relaxation to alleviate the antiaromaticity, lossing their energy and ultimately generating red emission. Despite single benzene fluorophores being the lightest molecules that exhibit red emission, they absorb blue light due to the vertical transition between the low-lying aromatic $S_0$ state and a high-lying $S_1$ Franck−Condon point, which could limit their utility in bioapplications. Thus, AF dyes signify a notable accomplishment by pushing the boundaries of the wavelength-to-molecular-weight ratio (WMR) inherent in conventional dyes.

An intriguing observation in AF dyes is the unusual connection between their rigid core structure and the rapid decay in the excited state. Our quantum chemical calculations on magnetic descriptors have revealed the aromaticity reversal between the $S_0$ and $S_1$ states, which leads to a strong vibration coupling within the antiaromatic fluorene skeleton and a high rate of non-radiative internal conversion. This emphasizes the importance of reducing vibrational coupling as a crucial step towards achieving brightly fluorescent AF dyes. Notably, the state-of-the-art non-adiabatic dynamics simulations, providing key atomistic details and dynamical features of the underlying internal conversion process for the studied dyes in the sub-picosecond timescale[58,59], have become a complementary and subsequent investigation. Impressively, the rapid decay in the excited state leads to ultrahigh photostability. AF dyes show good anti-photobleaching behavior under laser irradiation at power density of 200 kW cm⁻². This property may have potential applications in the field of single-molecule science that necessitates strong light-matter interaction. For example, single-molecule localization microscopy and Raman spectroscopy analysis always require high power density irradiation up to $10^4$-$10^6$ W cm⁻², [60,61]. In this context, it is worthwhile to explore additional single-molecule photophysical properties of AF dyes beyond fluorescence, such as Raman scattering, absorption, and photothermal and photoacoustic effects.

One benefit of the AF dyes is their high cell permeability. Currently, the focus of in vivo molecular imaging primarily centers on targeting extracellular structures and receptors rather than intracellular molecules and processes. Although there has been notable progress in the development of intracellular NIR-II nanosensors[62], further advancements in molecular technology with intracellular sensitivity hold the potential to introduce more diagnostic methods and shed light on unexplored biological questions. To this end, we have developed a proof-of-concept mini sensing platform, exemplified by AF2B detecting intracellular ROS production. AF2B restores cationic antiaromatic π-conjugation via amide breaking, generating a fluorogenic/acoustogenic signal response. This general activation mechanism can be applied to develop mini probes for monitoring intracellular calcium and metabolite dynamics, as well as enzyme catalytic activity.

AF dyes offer the additional benefit of sharply tuning their pharmacokinetics and biodistribution through simple chemical modifications. These factors directly influence clearance rate, biocompatibility, and in vivo imaging performance, including sensitivity, contrast, and diagnostic timeliness. In our study, a single carboxylation was demonstrated to alter the excretion pattern of AF2 and AF3 dyes, redirecting them from the hepatointestinal to the renal system. Remarkably, AF2-COOH and AF3-COOH, with molecular weights of 370 and 399 Da, respectively, represent the smallest contrast agents with efficient renal clearance. AF3-COOH exhibited advantages over the representative macromolecular contrast agent (43300 Da), advancing the diagnostic timeline by approximately 50 min in renal injury induced by ischemia-reperfusion. While the impact of other similar simple chemical modifications on the pharmacokinetics and biodistribution of AF dyes remains unknown, our findings demonstrate that much room remains for exploring the potential of mini dyes in bioimaging and clinical disease diagnosis.

In conclusion, a series of sterically protected antiaromatic AF dyes based on diaminofluorene skeleton with molecular weights of 299–504 Da and spectra covering 700–1600 nm were concisely synthesized. We have developed a mini molecular sensing platform based on antiaromatic skeleton, and showed the feasibility of mini AF skeleton for functionalization to access pharmacokinetics and biodistribution tunability, which made them powerful tools for in vivo near-infrared bioimaging and sensing. Overall, this study promotes the bioapplications of antiaromatic dyes and highlights antiaromaticity as a useful design rule of dye development to overcome the limitations of aromatic systems.

## Methods

### Animal handles
All animal procedures were performed in accordance with the guidelines of the Institutional Animal Care and Use Committee of Fudan University, in agreement with the institutional guidelines for animal handling. All of the animal experiments were authorized by the Shanghai Science and Technology Committee. Female Balb/c mice (4-6 weeks old, average weight of 20 g) were purchased from Shanghai JSJ Laboratory Animal Co. Ltd., randomly allocated form cages and housed in specific-pathogen free (SPF) and standard environmental conditions (22–25 °C, 45–55% humidity, 12/12 h dark/light cycle) with free access to water and fodder, and randomly selected from cages for all imaging experiments.

### Synthesis and general methods
For all synthesis, characterization, and additional details, please see Supplementary Information.

### Quantum chemical calculation
Unless otherwise noted, all density functional theory (DFT) and time dependent (TD)-DFT calculations were performed using Gaussian 16 code[63]. For atomic coordinates, see Supplementary Data 1.

1) Vertical ionization energy (VIE) of amino groups. To compute the vertical ionization energies of various amino groups, their corresponding amine structures (the fluorene group was replaced by methyl group) were firstly optimized using B3LYP[64]/6-311 G(d)[65–67]. Based on these optimized structures, we computed the electronic energies of ionized molecules using B3LYP/6-311 G(d), by removing one electron in each molecule. The differences between the electronic energies in their neutral and ionized states afford the vertical ionization energies of these amino groups.

2) Geometry optimization, excited state calculation and vibrational-resolved absorption spectra. The ground state ($S_0$) structures were optimized using density functional theory (DFT) at the B3LYP-D3BJ[68]/6-311 G(d,p) level with PCM(DCM)[69] solvent model where DFT-D3(BJ) dispersion correction were also applied. Frequency calculations were performed in all structural optimization to confirm that stable structures without imaginary vibrational frequencies were obtained. The vertical excitation energies for $S_1$ states were calculated using the TD-DFT method at the PCM(DCM)-LC-BLYP*/def-TZVP[70] level, where the symbol * represents the optimally-tuned range-separated (OTRS) parameter[71,72] based on the optimized $S_0$ structures. Note that such a OTRS functional method has been demonstrated to provide reliable predictions for the electronic structures and nuclear magnetic properties of molecular systems[73,74]. Vibrational-resolved absorption spectra were further calculated using Franck–Condon/Herzberg–Teller approximation (FCHT) approximation in Gaussian 16 program.

3) Aromaticity of AF dyes. The anisotropy of the induced current density (ACID) for AF dyes were calculated at based on the optimized $S_0$ structure using the ACID method[75] as implemented in Gaussian 16 program with keywords of (NMR = CSGT, Iop(10/93 = 1)). To obtain ACID result of $S_1$ state, the delta self-consistent field (ΔSCF) method[76,77] was applied to avoid NMR-unsupported TDDFT method. The orientation of the magnetic field is chosen to be orthogonal to the fluorene planes outward. The graphics were generated using the POV-Ray program. The nucleus-independent chemical shift scans along the X and Y axes (NICS-XY scan)[78,79] above the fluorene ring of AF dyes in the $S_0$ and $S_1$ states were calculated at the same level of theory, NICS(1)$_{zz}$ was defined as the zz component of the NICS value located 1 angstrom perpendicular to the plane above the fluorene ring. Besides, to exclude the method dependencies, three methods including DFT-B3LYP, DFT-LC-BLYP* and complete-active-space self-consistent field (CASSCF) method were used for AF3 calculation (Supplementary Table 6). Note that the CASSCF calculation is considered as the standard approach for studying the magnetic properties of $S_1$ excited state[35,80]. It can be seen that all the three methods give qualitatively consistent predictions for the NICS(1)$_{zz}$ values for both $S_0$ and $S_1$ states of AF3, supporting the main conclusion of this work. For multicenter index (MCI)[81,82] and electron localization function-pi (ELF-π)[83] calculation details, see Supplementary Note for Supplementary Fig. 22 in Supplementary Information file.

4) Reorganization energy, H-R factor and $k_r/k_{ic}$ of AF3. Reorganization energy and H-R factor of AF3 were derived based on the $S_0$ and $S_1$ optimized structures and the corresponding vibrational results were obtained by Dushin code[84]. The calculations of $k_r$ and $k_{ic}$ were performed at the PCM(DCM)-TD-B3LYP-D3(BJ)/def-TZVP level under 298.15 K using the MOMAP program[85].

### Measurement of chemical stability and selectivity
All dyes were dissolved in DMSO to obtain stock solutions of 20 mM. For measurement of chemostability in GSH, PBS (pH = 7.4) solution of GSH in different concentration was prepared ($10^{-4}$–$10^1$ mM). 2 μL of dye stock solution were added into GSH solution (2 mL) and transferred to a 4 mL quartz cell (1 cm optical length), then the absorption spectra of each dye were recorded in various concentrations of GSH solution once the absorbance reached a stable state. For measurement of response selectivity of AF2B, 50 eq. of GSH, $H_2O_2$, $\cdot O_2^-$, ClO$^-$ or 20 eq. of $\cdot$OH or 0.5 eq. of ONOO$^-$ was added to 50 μM AF2B solution in PBS, respectively, then absorption spectra were recorded after mixing for 1 min.

### Measurement of photostability in aqueous solution
AF2, AF3, AF11, AF14, and Cy7-Cl were dissolved in DMSO to obtain respective stock solution with concentration of 20 mM. A certain amount of the above stock solutions was dissolved in 2 mL 1×PBS and transferred to a 4 mL quartz cell (1 cm optical length). The absorbance at 808 nm (AF2, AF3, AF11, and Cy7-Cl) or 980 nm (AF14) was set to a same absorbance value (0.50). The solutions of AF2, AF3, and AF11 were illuminated with 808 nm laser at same power density of 1.6 W/cm$^2$ for 1 min, 4 min, 10 min, 20 min and 30 min. AF14 was illuminated with

980 nm laser at the same condition. For Cy7-Cl, the illumination was conducted with 808 nm laser at a short period of time (20 s, 40 s, 60 s, 80 s, 100 s, and 120 s for Cy7-Cl) due to photobleaching. The absorption spectra were measured after each illumination.

## Photostability comparison with live-cell imaging

Brain capillary endothelial cells (BCECs, catalog no.: PCLM0132-RT) were provided by Shanghai Bihe Biochemical Technology Co., Ltd. BCECs ($1 \times 10^5$) were cultured petri dish for 24 h. Then, medium was washed with 1×PBS and replaced with 1 μM AF3 and Cy7-Cl 1×PBS solution, respectively, and further incubated for 30 min under 37 °C. The above-treated groups were washed with 1×PBS twice, then fluorescent imaging was performed on an Olympus-IX71 epifluorescence microscope with excitation at 808 nm (200 kW cm$^{-2}$, exposure time: 100 ms per frame). Fluorescence images were collected with 1000LP.

## Cell permeability comparison by live-cell imaging and quantification

BCECs ($1 \times 10^5$) were cultured in petri dish for 24 h. Then, medium was washed with 1×PBS and replaced with AF3 (1 μM 1×PBS solution) and IR1048 (1 μM 1%DMSO in 1×PBS solution), respectively. IR1048 group were incubated for 30 min, AF3 groups were incubated for 30 s, 5 min and 10 min under 37 °C, and then washed with 1×PBS twice, respectively. Then fluorescent imaging was performed on an Olympus-IX71 epifluorescence microscope with excitation at 940 nm. Fluorescence images were collected with 1100LP for both AF3 and IR1048 groups.

CT26 (catalog no. TCM37) cells were provided by Cell Bank, Chinese Academy of Science. For cell permeability quantification of AF3 and IR1048, CT26 cells ($1.6 \times 10^7$) were suspended in 1 mL of 5 μM 1×PBS solution and incubated for 30 min under 37 °C, then cells were centrifuged to obtain supernatant liquid. The supernatant liquid was extracted with 1 mL of DCM (for IR1048 group, 10 μL Triton X100 was added to avoid emulsification), and UV–vis absorbance were measured together with 1 μM, 2 μM, 3 μM, 4 μM and 5 μM standard DCM solution (945 nm for AF3, 1048 nm for IR1048). The absorbances of standard solution were fitted to get calibration curve for the calculation of dye concentration in supernatant sample. The quantified cell permeability of each dye was calculated by the following formula:

$$P = \left(1 - \frac{c_{post}}{c_{pre}}\right) \times 100\% \tag{1}$$

Where $P$ is cell permeability, $c_{post}$ and $c_{pre}$ is the dye concentration before and after incubation with cells, respectively.

## ONOO⁻ response of AF2B in BCEC cells

BCECs ($1 \times 10^5$) were cultured in a 35 mm petri dish for 24 h. Then, the cell culture medium was replaced with 15 μM ONOO⁻ 1×PBS solution and incubated for 30 min. For control group, 400 μM uric acid was added as ONOO⁻ scavenger following the addition of 15 μM ONOO⁻ 1×PBS solution and the cells were incubated for 1.5 h. All groups were stained with 20 μM AF2B for 20 min, followed by washing with 1×PBS for 3 times to remove free probes. The fluorescence imaging was performed on an Olympus-IX71 epifluorescence microscope with excitation at 808 nm. Fluorescence images were collected with 900 LP.

## MSOT imaging and sensing of traumatic brain injury (TBI) mice model

TBI mouse model was established using the weight drop method according to the previous report[86]. Before establishing TBI model, mice were anesthetized with isoflurane and fixed on the stereotaxic apparatus, which padded with a heated surgical pad with stationary temperature of 37 °C. Following a sagittal incision, a craniotomy on the brain' s right hemisphere was conducted using a 4-mm-diameter trephine bur fitted to an electronic drill. Then, a 40-g metal bolt with an impact depth of 1.5 mm was dropped from a 20-cm height, which delivered a strike on the dorsal aspect of the skull (strike core was 3 mm right and 2 mm posterior of anterior fontanelle point) to induce the traumatic brain injury. Absorbance gelatin (Gelfoam, Pfizer) was applied to the trauma site to clean bleeding. Skin flaps were sutured together to close the wound. All mice were allowed to recover in a 37 °C heated surgical pad after injury and no mortality was found.

After the construction of TBI model for 12 h, mouse was intravenously injected with AF2B (200 μM, 200 μL 1×PBS solution), then PA imaging was conducted after 30 min. Images were captured by Vevo-LAZR (FujiFilm VisualSonics Inc.) system with Nd: YAG laser with optical parametric oscillator (OPO) (680 - 970 nm; 680 nm, 700 nm, 750 nm, 800 nm, 850 nm, 900 nm, 950 nm chosen for MSOT) as excitation source, PA gain value was properly set to minimize background noise interference. PA standard spectra of OxyHb and DeOxyHb were built-in in the system, spectra of AF2 were collect previously and set as reference spectra for MSOT.

## In-vivo wide-field whole-body NIR-II fluorescent imaging

AF2, AF3-COOH, AF2-COOH, and AF3-Dextran solutions were prepared (1 mM in 1×PBS solution). For AF2, 5% F127 (10% w/w DMSO solution) was added. The solutions were used for all animal imaging experiments. Female Balb/c mice were removed hair by shaving the body area, then hair removal cream was applied. After that, mice were intravenously injected with AF3, AF3-COOH, and AF3-Dextran (1 mM, 200 μL, 1×PBS solution), respectively. NIR-II fluorescence imaging was subsequently conducted under 940 nm excitation (180 mW cm$^{-2}$, 200 ms exposure time) and images were collected with 1000LP and 1100LP filter.

## In-vivo NIR-II fluorescent imaging for renal ischemia-reperfusion injury detection

AF3-COOH and AF3-Dextran solutions were prepared (1 mM in 1×PBS solution). The renal ischemia-reperfusion (RIR) mice model was established by surgical clamping of the renal pedicle of the right kidney (RK) for 30 min, then the clamps were removed and allowed reperfusion of RK for 30 min. AF3-COOH and AF3-Dextran were then intravenously injected to the RIR mice and NIR-II in vivo fluorescence imaging in the dorsal position was subsequently conducted under 940 nm excitation (180 mW cm$^{-2}$, 200 ms exposure time) and images were collected with 1000LP and 1100LP filter.

## In-vivo multispectral NIR-II fluorescent multi-organ functional imaging

Before animal imaging, the fluorescent signals of AF2 and AF3 solutions were collected respectively, under the excitation of 730 nm, 808 nm, and 940 nm lasers. The signals were plotted as excitation spectra, which were used for image unmixing via nonnegative matrix factorization (NMF) decomposition algorithm. For the multi-organ functional imaging experiment, mice were randomly divided into two groups. Mixed solutions of AF2/AF3-COOH and AF2-COOH/AF3-Dextran were respectively delivered using retro-orbital injection, and NIR-II fluorescent imaging was conducted 5 min postinjection. 730 nm, 808 nm, and 940 nm excitation were successively applied using the same condition as the experiment of excitation spectral acquisition. The collected image stacks were unmixed by imageJ software with a plugin of nonnegative matrix factorization (NMF) decomposition.

## Reporting summary

Further information on research design is available in the Nature Portfolio Reporting Summary linked to this article.

## Data availability

The data supporting the findings of this study are available within the paper and its Supplementary Information files and from the corresponding authors upon request. Source data are provided in this paper. Crystallographic data for the structures reported in this Article have been deposited at the Cambridge Crystallographic Data Centre, under deposition numbers CCDC 2263552 (AF3). Source data are provided in this paper.

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

## Acknowledgements

F.Z. acknowledges support from the National Key R&D Program of China (grant no. 2023YFB3507100), and the National Natural Science Foundation of China (NSFC, grant no. 22088101, 21725502, 51961145403), and the Research Program of Science and Technology Commission of Shanghai Municipality (grant no. 20JC1411700, 21142201000, 22JC1400400), and the Innovation Program of Shanghai Municipal Education Commission, and the New Cornerstone Science Foundation through the XPLORER PRIZE. S.W. acknowledges support from NSFC (grant no. 22004018, 22274030) and the Research Program of Science and Technology Commission of Shanghai Municipality (grant no. 23QA1407100). H.S. acknowledges support by NSFC (grant no. 12274128), the Research Program of Science and Technology Commission of Shanghai Municipality (grant no. 21QA1402600), and the ECNU Multifunctional Platform for Innovation (001) and HPC Research Computing Team for providing computational and storage resources and the support of the NYU-ECNU Center for Computational Chemistry at NYU Shanghai.

## Author contributions

F.Z., S.W., and H.S. supervised the research. F.Z. and S.W. conceived the research, and designed the experiments. S.W. and K.Y. conducted the synthesis, characterization and optical imaging experiments. K.Y. and Z-B.H. performed the quantum chemical calculation. K.Y., P.Y., Y.C., Z-Y.H. and J.C. contributed to the cell and animal experiments. F.Z., S.W. and K.Y. analyzed the results, and prepared the manuscript, Figures and supplementary information. All authors contributed to the discussion and editing of the manuscript.

## Competing interests

The authors declare no competing interests.
