## [Peer Review File · Nature Communications]

Ultra-photostable Small-molecule Dyes Facilitate Near-infrared BiophotonicsEditorial Note: Figure R3 in this Peer Review File has been amended to remove third-party material where no permission to publish could be obtained.

REVIEWER COMMENTS

Reviewer #1 (Remarks to the Author):

In this manuscript, the authors report the antiaromatic 4 π -electron five-membered ring-based dyes (AF) for NIR-II imaging. Overall, the manuscript is presented reasonably. However, this work did not show obvious advantages in the in-vivo imaging experiment (e.g., Figure 4) compared with previous reported probes. Additionally, some issues still need addressing:

1. The manuscript title “Ultra-photostable.....” should be modified. From the manuscript, it seems a bit hard to see the “ultra-photostable” feature of the dyes.
The stabilities of the dyes should be compared with typical commercially-available stable dyes.
2. The experiments proving the potentials of the dyes are too few, more mouse models should be employed to demonstrate the applicability of these dyes.
3. Table 1 shows that AF3 has the molecular weight of 355, and Figure S76 shows that Dextran-N3 has the molecular weight of 40000Da. But Figure 4 shows that AF3-Dextran has the molecular weight of 44000Da, which does not add up.
4. From Table 1, it is obvious that for the AF dyes, their Stokes shift is quite small and the fluorescence quantum yield is not high, which is no good for fluorescence imaging given the enhanced excitation-light interference and self-absorption. In addition, these dyes molar extinction coefficient is not high either, which is no good for photoacoustic imaging. These limitations would hinder the dyes’ practical applications, hence these limitations should be pointed in the manuscript.
5. In cell experiments, why choose AF3? It does not seem to have the best properties among all the dyes.
6. In Figure 2, photophysical mechanistic investigation was conducted for only AF3. This is inappropriate. The investigation should be conducted for other dyes.
7. In Figure 4, the number of repeating units of AF3-Dextran (x and y) should be clearly provided. AF3-Dextran should be properly characterized, e.g., NMR spectra and MALDI-TOF mass spectrum.
8. In Figure 5, the phenylboric acid group of AF2B would respond to various ROS members. The selectivity should be investigated.
The photoacoustic imaging quality is very poor. A larger area of the mice’s body should be included in the photoacoustic imaging, so that at least some anatomical features could be observed.
9. In Scheme 1a, which represents the AF dyes should clearly indicated.
In Scheme 1b, what NJ, FD and BTC stand for should be described in the figure captions.
The quality of scheme 1 should be improved.

10. The solvents used in the experiments (e.g., measurements of optical properties) should be described in the figure captions and the notes for the table.

Reviewer #2 (Remarks to the Author):

The authors introduced the concept of aromaticity reversal upon electronic excitation as a means of developing small gap emitter with relatively small molecular weights. Overall, the manuscripts contain significant materials that would support their main conclusions. However, I found that some more improvements are needed.

1) The ultimate theoretical approaches of studying the internal conversion processes are the location of conical intersections and nonadiabatic molecular dynamics simulations. Therefore, the conclusions regarding ultrafast dynamics need to be revised. Although their static calculations hint at the rapid dynamics, it would be much clearer if they could show how the dynamic events would occur in the atomistic details in the sub-picosecond timescale. The experimental timescale of 7.41 ps only represents the slow dynamics after the initial rapid motions.

2) This is especially important that their conclusions are based on the rapid internal conversions. Some nearby states such as S₂ can be also involved during the processes.

3) The concept of an antiaromaticity design principle for low band gap with small molecular weight has already been demonstrated in their ref. 12, which needs to be properly credited. The authors need to be clear about what exactly is their original idea.

Reviewer #3 (Remarks to the Author):

Yan et al explored a new set of fluorophores which when compared to earlier ones are smaller and this brings a number of benefits. As the unique approach in their molecular design they make use of ground state antiaromaticity/excited state aromaticity. Yet, it is the computational investigations of the aromaticity effects that are the weakest in the manuscript; far below the quality one can expect to find in a paper published in a high-impact journal.

The aromaticity assessment is based on magnetic (NICS and ACID) and geometric aromaticity aspects (changes in BLA), yet this analysis is very unconvincing. First, it is written that NICS and ACID were calculated for the S₁ state, however, these computations were apparently done with (what the authors call) the DeltaSCF method. No reference is given to this method, but I suspect that it is the approach which was used by Kawashima et al in <https://onlinelibrary.wiley.com/doi/10.1002/anie.201904882>, and heavily criticized by Karadakov and Saito in

<https://onlinelibrary.wiley.com/doi/full/10.1002/anie.202001934>. That computational approach is easy but should never be used as the state which results is not a pure singlet state but a mix of singlet and triplet multiplicity. Much more useful would be a small CASSCF. Of course, the latter will not provide the

authors with an ACID plot, but they can calculate NICS at CASSCF level with the Dalton program.

With regard to the ACID plots in Figure S7 they are of so exceptionally low resolution that even if I magnify to 500% I cannot view the small current density arrows generated by the AICD program and which represent the alleged ring currents. The present plots are totally meaningless and indicate that the authors are unfamiliar with what is required for a proper (anti)aromaticity assessment.

Coming to the NICS data, it has been clarified numerous times that single NICS values at specific points in space should never be used for polycyclic molecules. Instead one should use NICS-XY scans as clearly described by Gershoni-Poranne and Stanger in <https://chemistry-europe.onlinelibrary.wiley.com/doi/full/10.1002/chem.201304307> The present NICS data should be recalculated with that method.

The analysis of the BLA values is odd; on page 5 of the manuscript is written that the BLA is reduced to 1.374 – 1.463 Å, however, it does not tell from what range. Also, the CC bond length range 1.374 – 1.463 Å is quite large and not indicative of extensive Baird-aromaticity in the S1 state. Indeed, one can see in Figures 2e and 2f that the reduction in BLA upon excitation from S0 to S1 is rather modest.

Far too often do aromaticity analyses by experimentalists primarily rely on magnetic aromaticity indicators, however, there is a growing body of studies that show the difference between magnetic indices, on the one side, and electronic and energetic indices on the other (see e.g. <https://chemistry-europe.onlinelibrary.wiley.com/doi/full/10.1002/slct.201602080>). Thus, the authors should confirm the switch in aromatic and antiaromatic character upon excitation by some electronic index (e.g. MCI) which can be applied to both S0 and S1 (the latter at TD-DFT level).

The references on the excited state aromaticity are not properly selected and placed. The Chem Rev by Rosenberg et al on excited state (anti)aromaticity (ref 20) is used as a reference for Hückel-aromaticity in the ground state, and the purely computational study by Karadakov (ref 31) is used to support that it “has been experimentally proved that [Baird’s rule] likewise applies to the lowest singlet excited state”. It is obvious that the authors have not read the literature on excited state aromaticity properly. For experimental studies which support that Baird’s rule can be applied to the lowest singlet excited states I propose to cite <https://pubs.acs.org/doi/10.1021/ja00060a063> (together with <https://chemistry-europe.onlinelibrary.wiley.com/doi/full/10.1002/cplu.201900066>), <https://www.nature.com/articles/s41467-017-00382-1> and <https://pubs.acs.org/doi/full/10.1021/jacs.0c05611>. In addition to the Chem Rev by Rosenberg I propose to add the recent review by Yan et al (<https://chemistry-europe.onlinelibrary.wiley.com/doi/full/10.1002/chem.202203748>). The computational paper by Karadakov (ref 31) should definitely be given, yet I would also add <https://pubs.rsc.org/en/content/articlelanding/2011/cp/c1cp22239b> and <https://pubs.acs.org/doi/10.1021/acs.joc.6b02460>. I also think they should cite Peter Wan’s pioneering experimental work on photosolvolytic to fluorenyl cations, see <https://pubs.rsc.org/en/content/articlelanding/1985/c3/c39850001207>

Finally, I wonder why they have used different functionals for different properties? The orbitals and orbital energies in Figure S3 were calculated with LC-BLYP* while those in Scheme 1 by B3LYP.

Summarized, the quality of the (excited state) aromaticity analysis is far below what can be expected from a manuscript submitted to a high-impact journal such as Nat. Commun. The aromaticity analysis is very speculative and insufficiently supported. There are many more items to address also, yet my time is simply limited. Those items will need to be addressed in further potential revisions.

Reviewer #1 (Remarks to the Author):

General comment: In this manuscript, the authors report the antiaromatic 4pi-electron five-membered ring-based dyes (AF) for NIR-II imaging. Overall, the manuscript is presented reasonably. However, this work did not show obvious advantages in the in-vivo imaging experiment (e.g., Figure 4) compared with previous reported probes. Additionally, some issues still need addressing:

Response: We sincerely appreciate the reviewer for her/his valuable comments and suggestions that have improved the manuscript. In response, we conducted a thorough revision, with a specific focus on illustrating the advantages of AF dyes in a mouse model of renal ischemia-reperfusion. Our findings demonstrated that AF3 can swiftly and distinctly detect renal injury, surpassing the performance of existing renal-clearable macromolecular probes. This advancement accelerates the diagnostic timeline of renal injury by approximately 50 minutes, as detailed in “response to comment 2”.

Furthermore, we have incorporated new data to substantiate the ultra-photostable property of AF dyes (response to comment 1). Additionally, the discussion section has undergone comprehensive revision to underscore the major implications derived from this study. We are confident that these substantial revisions effectively address the concerns raised by the reviewer.

Comment: 1) The manuscript title “Ultra-photostable....” should be modified. From the manuscript, it seems a bit hard to see the “ultra-photostable” feature of the dyes.

The stabilities of the dyes should be compared with typical commercially-available stable dyes.

Response: We appreciate the reviewer for the comments. In the previous submission, we have demonstrated that both AF3 and IR1048 showed no photobleaching in solution, outperforming Cy7 (Figure S6, the data of IR1048 was not shown). To further prove the ultra-photostable feature of our AF dyes, in this revision, we have included a comparison of the photostability of AF3 and IR1048 via a microscope assay. This assay adopted a strict condition, including depositing the dyes onto a slide, exposing them to air, and irradiating with intense laser (200 kW cm^{-2}) for 2 hours. As shown in the revised Figure S6c and S6d, the dyes displayed punctate signals in the slides, and notably, we still did not observe photobleaching of AF3, whereas IR1048 exhibited over 50% photobleaching during

this time. It is worth mentioning that, in an independent experiment previously performed by our lab, a D-A-D-type dye, TPB, was completely photobleached in only 20 minutes using the same condition (Figure R1). Both IR1048 and TPB have been recognized as photostable dyes in previous literatures, such as *Adv. Mater.* 2022, 34, 2206765; *Small* 2021, 17, 2101397; *Nat. Mater.* 2015, 15, 235.

We have made the following revisions in the manuscript and supplementary information:

Page 27 in the supplementary information, Figure S6 was updated:

Figure S6. (a) Photostability comparison of AF2, AF3, AF11, Cy7-Cl (808 nm ex), and AF14 (980 nm ex) in aqueous solution. The absorbance of AF2, AF3, AF11, Cy7-Cl at 808 nm and AF14 at 980 nm were set to 0.5 and laser power density was 1.6 W/cm^2 . (b) HPLC chromatogram of AF dyes before and after irradiation. (c, d) NIR-II fluorescence microscopic images of dyes exposed to air after certain time irradiation under 808 nm (AF3) and 980 nm (IR1048) laser excitation (d), and fluorescence intensity statistic results (c). $40\times/0.95$ objective was used. Laser power density: 200 kW cm^{-2} . Samples were prepared using microscope slides by dropping DCM solution of each dye and slowly evaporating.

Figure R1. Photostability comparison of AF3 and TPB. Chemical structure of TPB were also shown (i.e., fluorophore skeleton of CH1055).

Page 8 in the main text, this paragraph was rewritten:

Photobleaching presents a major obstacle when revisiting staining samples. Therefore, we conducted in vitro light exposure assay to examine the photostability of AF dyes. In aqueous solution, AF dyes (AF2, AF3, AF11, AF14) showed no changes in absorbance or chemical structure after 30 minutes of laser exposure (808 nm for AF2, AF3, AF11, and 980 nm for AF14 with power density of 1.6 W cm^{-2}), while Cy7, a commonly used NIR dye experienced rapid absorbance attenuation within 3 minutes of exposure to 808 nm laser at the same power density (Figure S6). However, with the same light exposure, we did not observe photobleaching of IR1048 in organic solvents (IR1048 is not water-soluble), which may be attributed to the low band gap of NIR-II dye minimizes the production of photooxidative species. For a thorough comparison, AF3 and IR1048 were subjected to a microscope assay in which the dyes were deposited onto a slide, exposed to air, received intense laser irradiation (808 nm for AF3 and 980 nm for IR1048 with power density of 200 kW cm^{-2}) and simultaneously recorded the fluorescence changes. It was observed that after 2 hours of irradiation, AF3 retained nearly 90% of its fluorescence signals, whereas IR1048 exhibited a decay exceeding 50% (Figure S6). The exceptional photostability of AF3 was further demonstrated in continuous cell imaging, allowing for over 1000 repetitions of snapshot recording with intense 808 nm laser irradiation (200 kW cm^{-2}). Notably, the images kept relatively high and stable SBR (Figure 3d, Supplementary Video 1). In sharp contrast, when Cy7-stained cells were repetitively excited with 808 nm at the same power density, they underwent complete bleaching in fewer than 15 frames (Figure 3e, Supplementary Video 2). Overall, these results highlight the ultra-photostability of AF dyes, which hold significant promise for optical imaging in challenging conditions.

Comment: 2) The experiments proving the potentials of the dyes are too few, more mouse models should be employed to demonstrate the applicability of these dyes.

Response: We appreciate the reviewer for the comments. In this revision, we have added an experiment to illustrate the advantages of AF dyes in the diagnostic of renal injury. Using a mouse model of renal ischemia-reperfusion, we demonstrated that AF3-COOH exhibited superiority over

the representative macromolecular renal-clearance contrast agent (43300 Da), advancing the diagnostic timeline by approximately 50 minutes. We made the following revision to the manuscript:

Page 9-10 in the main text, the following paragraph has been added:

Inspired by its rapid renal clearance feature, we were driven to investigate the potential applications of AF3-COOH in pre-clinical diagnosis. For example, renal ischemia-reperfusion (RIR) is a critical clinically acute renal injury with high mortality, and currently there is a lack of reasonable surveillance techniques for the rapid and early detection of RIR.^{52, 53} Previous studies have employed the clearance hindrance of water-soluble fluorescent macromolecules in injury kidney for optical detection of RIR, but it is time-consuming to reach a high SBR for accurate diagnosis due to their slow renal clearance.⁵⁴ With the anticipation that AF3-COOH could potentially rapidly diagnose RIR, we intravenously injected AF3-COOH in a mouse model (Figures 4g-i), featuring RIR in the right kidney (RK) and a healthy left kidney (LK). The same procedure was applied to AF3-dextran, serving as a macromolecular control. It was observed that the fluorescence signals of AF3-COOH were rapidly cleared from the healthy kidney in 11 minutes, whereas this process took 60 minutes for AF3-dextran (Figure 4h, S25). Conversely, both agents gradually accumulated in the RIR kidney, resulting in enhanced fluorescence signals during the 60 minutes following injection. Therefore, AF3-COOH achieved a SBR (defined as the RK/LK ratio) surpassing the Rose criterion ($SBR > 5$) for 100% certainty⁷ within 11 minutes, which outperformed AF3-dextran by sixfold in speed (Figure 4i). These results demonstrate the great value of AF dyes in clinical diagnosis, with the potential to advance diagnosis time and increase biocompatibility.

Page 19 in the method, the following paragraph has been added:

In-vivo NIR-II fluorescent imaging for renal ischemia-reperfusion injury detection. AF3-COOH and AF3-Dextran solutions were prepared (1 mM in 1×PBS solution). Renal ischemia-reperfusion (RIR) mice model was established by surgical clamping of the renal pedicle of the right kidney (RK) for 30 min, then the clamps were removed and allowed reperfusion of RK for 30 min. AF3-COOH and AF3-Dextran were then intravenously injected to the RIR mice and NIR-II in vivo fluorescence imaging in the dorsal position was subsequently conducted under 940 nm excitation (180 mW cm⁻², 200 ms exposure time) and images were collected with 1000LP and 1100LP filter.

New data has been added to the Figure 4 and Figure S25:

Figure 4 | Functionalized AF dyes for in-vivo fluorescent imaging. ... **g**, Schematic illustration of the ischemia-reperfusion process and in vivo NIR-II imaging. Right kidney was 30 minutes ischemia treated. **h**, In vivo NIR-II fluorescent images of renal ischemia-reperfusion (RIR) mice from the dorsal position at different time points after intravenous injection of each probe. The dashed circles indicate the kidney position. The bright fluorescence signals near the left kidney of AF3-COOH-treated group were mainly from liver (1 to 7 minutes) and intestine (20 to 80 minutes). **i**, Time-dependent evolution of the fluorescence intensity ratio of right-to-left kidney (RK/LK) for the images in **h**. The data are the mean \pm s.d. derived from $n = 3$ biologically independent mice. ...

Figure S25. (a) In vivo NIR-II fluorescent images of control group mice without renal ischemia-reperfusion treatment. The yellow circles indicate the kidney position. (b) Time-dependent evolution of the fluorescence intensity ratio of right-to-left kidney (RK/LK) for images in (a).

Comment: 3) Table 1 shows that AF3 has the molecular weight of 355, and Figure S76 shows that Dextran-N3 has the molecular weight of 40000Da. But Figure 4 shows that AF3-Dextran has the molecular weight of 44000Da, which does not add up.

Response: We appreciate the reviewer for identifying inaccuracies in our manuscript. The molecular

weight of AF3-dextran was determined by NMR and UV-Vis characterization, as detailed below:

1) Based on the NMR result (Figure S91), the average number of azido groups in each Dextran-N3 molecule can be calculated as 11.97. Each azido group increases the molecular weight by approximately 126 Da ($C_4H_6ON_4$). Therefore, the average molecular weight of Dextran-N3 is $40000 + 1508 = 41508$ Da.

2) Based on the UV-Vis absorption spectroscopy, the average number of AF3 in each AF3-Dextran molecule can be calculated as 4.7. Each AF3 increases the molecular weight by approximately 379 Da. Therefore, the average molecule weight of AF3-Dextran is $41508 + (379 * 4.7) = 43289 \approx 43300$ Da.

We made the following revision to the manuscript:

Page 9 in the main text:

For a direct comparison, we synthesized an AF3-dextran conjugate with a molecular weight of approximately **43300 Da** (details shown in supplementary information), which represents a conventional water-soluble modification approach for large-structure dyes.

Figure 4a, 4b:

Figure 4 | Functionalized AF dyes for in-vivo fluorescent imaging. a, Chemical structures of AF2-COOH, AF3-COOH and AF3-Dextran. **b,** Molecular weight, net charge and cLog D of modified and unmodified AF dyes.

cLog D was calculated using ChemAxon software. ...

Comment: 4) From Table 1, it is obvious that for the AF dyes, their Stokes shift is quite small and the fluorescence quantum yield is not high, which is no good for fluorescence imaging given the enhanced excitation-light interference and self-absorption. In addition, these dyes molar extinction coefficient is not high either, which is no good for photoacoustic imaging. These limitations would hinder the dyes' practical applications, hence these limitations should be pointed in the manuscript.

Response: We appreciate the reviewer for the critical comments.

First, we would like to clarify that the Stokes shifts of AF dyes ranges between 50 to 200 nm, which at least align with the majority of existing NIR-II dyes (generally less than 50 nm, such as *Angew. Chem. Int. Ed.* 2021, 60, 16294). We acknowledge that AF dyes exhibit relatively lower quantum yields compared to the best NIR-II dyes, however, their quantum yields still align with some D-A-D-type, or rhodamine dyes that have been used for high-contrast angiography, such as CH1055 (0.03% in water, *Nat. Mater.* 2015, 15, 235), EB766 (0.01% in water, *Nat. Mater.* 2021, 20, 1571), and VIX-4 (0.004% in water, *J. Am. Chem. Soc.* 2021, 143, 17136). The molar extinction coefficients of AF dyes range between 20000 to 40000 M⁻¹ cm⁻¹, which are higher than most of the D-A-D dyes that have been used for photoacoustic imaging, such as BBTT (6830 M⁻¹ cm⁻¹, *Chin. Chem. Lett.* 2021, 32, 1580-1585), CH1000 (8283 M⁻¹ cm⁻¹, *ACS Nano* 2017, 11, 12, 12276–12291), and BAF4 (16300 M⁻¹ cm⁻¹, *Angew. Chem. Int. Ed.* 2021, 60, 22376-22384).

In addition, we would like to highlight that the most valuable property of AF dyes is their miniaturized structure. We revised the discussion section in the main text.

Comment: 5) In cell experiments, why choose AF3? It does not seem to have the best properties among all the dyes.

Response: We appreciate the reviewer for the comments. Apart from AF3, we also conducted cellular imaging experiments with other two representative structures, AF2 and AF14 (**Figure S24**). All dyes show very high cell permeability. Indeed, AF2-stained cell appeared brighter than AF3, presumably due to its smaller molecular weight and higher brightness. However, for the unity of the discussion, we only present the data of AF3 in the main text.

Figure S24 in Supplementary Information:

Figure S24. Fluorescence images of AF dye-stained endothelial cells (ECs). Dye concentration: 1 μM in 1 \times PBS solution. Staining time: 10 min. Fluorescence images were captured under 808 nm excitation and 1000 LP filter.

Comment: 6) In Figure 2, photophysical mechanistic investigation was conducted for only AF3. This is inappropriate. The investigation should be conducted for other dyes.

Response: We appreciate the reviewer for the comments. Regarding the quantum yield, we reasoned that it is the fluorene skeleton, not their N-substituent groups, determining the quantum yields, as all AF dyes exhibits similar values (Table 1). In the revised manuscript, we have conducted aromaticity calculations for all dyes (Figure S7-S21). These results have driven the conclusion in Page 6 of the main text that “for all AF dyes, both nucleus-independent chemical shift scans along the X and Y axes (NICS-XY scan) and calculations of anisotropy of the induced current density (AICD) indicate a highly localized antiaromaticity in the five-member ring of the S_0 state, coupled with increased aromaticity in the entire fluorene skeleton of the S_1 state”. Combined with the structural inspection and non-radiative decay analysis, these static calculations have revealed the aromaticity reversal between the S_0 and S_1 states, which leads to a strong vibration coupling within the antiaromatic fluorene skeleton and a high rate of non-radiative internal conversion.

Comment: 7) In Figure 4, the number of repeating units of AF3-Dextran (x and y) should be clearly provided. AF3-Dextran should be properly characterized, e.g., NMR spectra and MALDI-TOF mass spectrum.

Response: We appreciate the reviewer for the comments. We attempted to characterize AF3-dextran using NMR and Maldi-TOF techniques, but unfortunately, NMR only can prove the presence of AF3

conjugated to dextran (**Figure R2**) and Maldi-TOF did not achieve the intended results. Therefore, we calculated the number of AF3 conjugated to dextran using the UV-Vis absorption method. Regarding the characterization and calculation of AF3-Dextran, please see our response to comment #3.

Figure R2. NMR spectrum of AF3-Dextran.

We revised the structure of AF3-dextran in Figure 4a, as shown in response to comment 3.

Comment: 8) In Figure 5, the phenylboric acid group of AF2B would respond to various ROS members. The selectivity should be investigated.

The photoacoustic imaging quality is very poor. A larger area of the mice's body should be included in the photoacoustic imaging, so that at least some anatomical features could be observed.

Response: We appreciate the reviewer for the comments. We have presented the ROS reaction selectivity of AF2B (**shown in Figure S29**), which indicates that AF2B selectively react with ONOO⁻, and cannot be activated by other ROS species.

Regarding the imaging quality, it is primarily determined by the imaging equipment we used

(VevoLAZR, FujiFilm VisualSonics Inc.), which operates on the principles of tomography. It is well known that photoacoustic tomography has lower spatial resolution than photoacoustic microscopy (*Science* 2012, 335, 1458-1462). For example, we compared some of the photoacoustic results from previous published articles that utilized the same instrument (e.g., figures in *Theranostics* 2015, 5(3), 289–301 and *ACS Nano* 2015, 9(10), 9517-9527, **shown in Figure R3**). The consistent image quality between these studies and our own strongly underscores the detection reliability of the AF2B probe in the TBI model. We believe that AF dyes, as a superior acoustogenic platform, can be better demonstrated in advanced photoacoustic facilities.

Regarding the anatomical features in the photoacoustic images, due to the small size of the mouse brain as the subject, we only can see the boundary between the skull and brain parenchyma. To our knowledge, this limitation in resolution is inherent to photoacoustic tomography.

Figure S29. Selectivity of AF2B over ClO[·] (50 eq.), H₂O₂ (50 eq.), ·OH (20 eq.), ·O₂[·] (50 eq.), GSH (50 eq.) and ONOO[·] (0.5 eq.).

[REDACTED]

Figure R3. Photoacoustic tomography images in referenced literatures and this work.

Comment: 9) In Scheme 1a, which represents the AF dyes should clearly indicated.

In Scheme 1b, what NJ, FD and BTC stand for should be described in the figure captions.

The quality of scheme 1 should be improved.

Response: We appreciate the reviewer for the comments. In the revised manuscript, we have made necessary modifications to Scheme 1 and the relevant paragraph (*shown below*) to make our description clearer.

Page 2 in the main text:

..... Common dye skeletons mostly display $[4n+2]\pi$ -electron aromatic character (Scheme 1a left).....

Page 3 in the main text:

..... However, those approaches make dye skeletons larger (Scheme 1b, S2).....

..... One may expect a naturally reduced HOMO-LUMO gap in a $[4n]\pi$ -electron system because ground-state antiaromaticity energetically raises HOMO while excited-state aromaticity decreases LUMO (Scheme 1a right).....

Scheme 1 | Concept of antiaromatic dyes, and comparison between AF dyes and conventional dyes. **a**, The concept description of heteroatom-free antiaromatic molecular dyes. **Left: common dyes with ground-state aromaticity, right: AF dyes with ground-state antiaromaticity.** **b**, Graphic showing the relationship between molecular weight (MW) and wavelength of different types of fluorescent dyes. All dyes were numbered, and their corresponding chemical structures and properties were shown in Scheme S2 and Table S1. **Characters such as FD1080 and BTC1070 were adopted from their names in previous literatures.**

Comment: 10) The solvents used in the experiments (e.g., measurements of optical properties) should be described in the figure captions and the notes for the table.

Response: We appreciate the reviewer for the comments. We have conducted a thorough review and made the necessary additions accordingly. Relevant corrections were *shown below*.

Table 1 | Photophysical properties of AF Dyes in DCM.

Dye	Molecular weight ^a (g/mol)	λ_{abs}^b (nm)	λ_{em}^b (nm)	ϵ^c (M ⁻¹ cm ⁻¹)	Peak-shoulder ratio	Stokes Shift (nm/cm ⁻¹)	Quantum Yield (10 ⁻² %)
AF1	299	800	976	9047	0.629	176/2254	0.3
AF2	327	847	933	14869	0.987	86/1088	2.3
AF3	355	959	1014	23052	1.189	55/565	1.1
AF4	387	983	1046	19795	ND ^d	63/613	0.6
AF5	411	966	1015	25261	1.274	49/500	1.3
AF6	379	962	1014	21685	1.184	52/533	1.3
AF7	451	946	995	15231	1.050	49/521	1.1
AF8	407	968	1017	24705	1.291	49/498	1.3
AF9	435	966	1026	26793	1.258	60/605	1.4
AF10	463	973	1026	22143	1.345	53/531	1.2
AF11	495	994	1045	21081	ND ^d	51/491	1.2
AF12	479	965	1029	18205	1.273	64/645	ND ^d
AF13	603	1045	1196	30777	1.476	151/1208	0.5
AF14	503	1101	1187	41132	1.933	86/658	0.8

^aMolecular weight of AF dyes excluding counter ion, ^babsorption and emission maxima, ^cMolar absorptivity, ^dND=not determined.

Figure 1 | Synthesis, spectral characteristics and structure-property relationships of AF dyes. a, b, Two-step synthetic route for AF1~AF14 (**a**) and their corresponding substituent patterns (**b**) with graphic showing the absorption maxima on a spectrum axis. Representative spectrally-distinctive dyes (AF2, AF3, AF11 and AF14) were shown in colored solid lines. **c, d,** Normalized absorption (**c**) and emission (**d**) spectra of AF2, AF3, AF11 and AF14. **e, f,** Plots indicating the linear relationship (No. 1, 2, 8, 11 and 12 were excluded from linear fitting) between absorption maxima (**e**)/peak-shoulder ratio (**f**) of AF dyes and calculated vertical ionization energies of amine substituents. All spectral properties were collected in dichloromethane.

Figure 2 | Photophysical mechanistic investigation of AF3. a, b, (a) fs-TA spectrum of AF3 and (b) single-wavelength kinetic fit result at 977 nm. Solvent: dichloromethane. **c,** Jablonski diagram of AF3 excitation process. **d,** X-ray crystal structure of AF3. **e, f,** AF3 bond length comparison of X-ray structure and S₀/S₁ optimized structures on the central 5-membered ring and the whole fluorene skeleton shows strong BLA. **g-i,** (g) Calculated reorganization energy, (h) electronic coupling factor, and (i) k_{ic} versus normal vibration mode wave numbers for S₁ state of AF3. Representative normal modes were shown as insets.

Figure 5 | Mini sensing platform based on AF2 for in-vivo ONOO⁻ detection. a, Chemical structures of AF2Ac and AF2B. **b,** Variation in absorption spectra of AF2B (50 μ M) in PBS pH 7.4 after incubation with increasing concentration of ONOO⁻ (0-25 μ M, gradient: 5 μ M). **c,** PA spectra of AF2B (50 μ M) in PBS pH 7.4 before and after

the addition of 25 μM ONOO⁻. **d**, Fluorescence images of AF2B-stained brain capillary endothelial cells (BCECs) pre-treated with a vehicle control (Blank), or 1 μM ONOO⁻, or 1 μM ONOO⁻ + 400 μM uric acid (UA). The bars represent mean \pm s.d. derived from n = 20 independent cells. Statistical analysis was performed using two-tailed Student's t-test ($\alpha = 0.05$), ***: p < 0.001. **e**, Schematic illustration of the construction of TBI mice model and the timeline for probe administration and MSOT imaging. **f**, MSOT/US images of OxyHb, DeoxyHb and AF2 in the left (normal) and right (TBI) hemisphere of a same mouse after i.v. injection of AF2B. The bars represent mean \pm s.d. derived from n = 3 biologically independent mices.. Statistical analysis was performed using two-tailed Student's t-test ($\alpha = 0.05$), ***: p < 0.001.

Reviewer #2 (Remarks to the Author):

General comment: The authors introduced the concept of aromaticity reversal upon electronic excitation as a means of developing small gap emitter with relatively small molecular weights. Overall, the manuscripts contain significant materials that would support their main conclusions. However, I found that some more improvements are needed.

Response: We appreciate the reviewer for the positive comments and valuable suggestions. In response, we conducted a comprehensive revision, which we believe has significantly improved this manuscript.

Comment: 1) The ultimate theoretical approaches of studying the internal conversion processes are the location of conical intersections and nonadiabatic molecular dynamics simulations. Therefore, the conclusions regarding ultrafast dynamics need to be revised. Although their static calculations hint at the rapid dynamics, it would be much clearer if they could show how the dynamic events would occur in the atomistic details in the sub-picosecond timescale. The experimental timescale of 7.41 ps only represents the slow dynamics after the initial rapid motions.

Response: We appreciate the reviewer for the comments. Concerning the usage of the terms "rapid" and "ultrafast" in our manuscript, we intentionally employed them to highlight the accelerated dynamics of AF dyes in transitioning from the S₁ state to the S₀ state, aiming to differentiate them from existing NIR-II dyes, rather than indicating hyperfine dynamic processes after excitation. We apologized for any confusion that may have arisen from this choice of terminology.

Regarding the theoretical approach employed in this work, it is well-established that the related molecular fluorescence properties regarding the radiative and nonradiative processes can be quantitatively obtained from first-principle calculations coupled with correlation function formalism for the internal conversion under harmonic approximation by considering the Duschinsky rotation effect [Shuai et al. *J. Am. Chem. Soc.* 2007, 129, 9333; *Acc. Chem. Res.* 2014, 47, 3301]. As shown in Figure 2i, the internal conversion rate (k_{ic}) of each promoting mode, and a calculated total k_{ic} ($1.06 \times 10^{11} \text{ s}^{-1}$) agrees well with the experimental measurement ($1.4 \times 10^{11} \text{ s}^{-1}$). Notably, our main finding suggests the total contribution of C-C stretching mode to k_{ic} is much greater than that of the

high-frequency C-H stretching mode. Alternatively, we totally agree that the state-of-the-art non-adiabatic dynamics simulations can provide a robust tool to reveal the atomistic details in the sub-picosecond timescale and produce key dynamical features of the underlying internal conversion process for the studied dyes [Martínez et al. *Chem. Sci.* 2022, 13, 373; *J. Am. Chem. Soc.* 2022, 144, 28, 12732]. We hope that the reviewer can understand that the expensive and technical non-adiabatic dynamics simulations are beyond the research capabilities of our current theoretical research group. However, we may reserve this valuable suggestion using non-adiabatic dynamics simulations as a subsequent study in the future.

Accordingly, in the revised discussion section, we add the comments “Notably, the state-of-the-art non-adiabatic dynamics simulations, providing key atomistic details and dynamical features of the underlying internal conversion process for the studied dyes in the sub-picosecond timescale [Martínez et al. *Chem. Sci.* 2022, 13, 373; *J. Am. Chem. Soc.* 2022, 144, 28, 12732], have become a complementary and subsequent investigation.”.

Comment: 2) This is especially important that their conclusions are based on the rapid internal conversions. Some nearby states such as S₂ can be also involved during the processes.

Response: We appreciate the reviewer for the comments, however, the nearby excited states such as S₂ should not be involved during the processes. In this study, we employed the laser excitation at 730 nm, 808 nm, and 940 nm for all experiments related to photophysical properties. These wavelengths correspond to the S₀→S₁ transition, as both UV-Vis absorption spectra (Figure S2) and theoretical calculations (**Table R1**) confirmed that the wavelengths corresponding to S₀→S₂ transitions for all AF dyes locate around 400-500 nm.

Table R1. Theoretical calculation result of excitation energies of AF dyes.

Dye	$\Delta E_{S_0-S_1}/\text{eV}$	$\Delta E_{S_0-S_1}/\text{nm}$	$\Delta E_{S_0-S_2}/\text{eV}$	$\Delta E_{S_0-S_2}/\text{nm}$
AF1	1.656	749	2.913	426
AF2	1.698	730	2.909	426
AF3	1.565	792	2.835	437
AF11	1.476	840	2.370	523
AF12	1.488	833	2.724	455

AF13	1.383	897	2.431	510
AF14	1.353	917	2.471	501

Comment: 3) The concept of an antiaromaticity design principle for low band gap with small molecular weight has already been demonstrated in their ref. 12, which needs to be properly credited. The authors need to be clear about what exactly is their original idea.

Response: We appreciate the reviewer for the comments. We apologize for any ambiguity in explaining the design principle of AF dyes in our manuscript. Basically, the main difference on the design principle is that our AF dyes belong to a type of **ground-state antiaromatic** molecules, whereas in ref. 12, their molecules belong to **excited-state antiaromatic** molecules (illustrated differences *in Scheme 1a*). We have discussed this difference in the revised discussion section, as shown below:

“In the present work, we have developed a series of AF dyes with a $[4n]\pi$ -electron Hückel antiaromatic skeleton, achieving an increased WMR of 2~2.7 ($\lambda_{\text{abs}}/\text{Mw}$), surpassing that of most of conventional dyes (WMR < 2) (Table S1). The AF skeleton exhibits ground-state antiaromaticity and excited-state aromaticity, which is opposite from conventional dyes with $[4n+2]\pi$ -electron Hückel aromatic character. This unique characteristic results in an inherently narrow HOMO-LUMO gap, with the destabilized antiaromatic S_0 state elevating the HOMO energy while the stabilized aromatic S_1 state reduces the LUMO energy. It is worth noting that the influence of ground-state antiaromaticity on wavelength differs from the S_1 antiaromaticity relief mechanism recently disclosed in single benzene fluorophores.¹² The latter are aromatic in their S_0 and antiaromatic in their S_1 . Their S_1 states undergo geometry relaxation to alleviate the antiaromaticity, losing their energy and ultimately generating red emission. Despite single benzene fluorophores being the lightest molecules that exhibit red emission, they absorb blue light due to the vertical transition between the low-lying aromatic S_0 state and a high-lying S_1 Franck–Condon point, which could limit their utility in bioapplications. Thus, AF dyes signify a notable accomplishment by pushing the boundaries of the wavelength-to-molecular-weight ratio (WMR) inherent in conventional dyes.”

Scheme 1 | Concept of antiaromatic dyes, and comparison between AF dyes and conventional dyes. **a**, The concept description of heteroatom-free antiaromatic molecular dyes. **Left: common dyes with ground-state aromaticity, right: AF dyes with ground-state antiaromaticity.** **b**, Graphic showing the relationship between molecular weight (MW) and wavelength of different types of fluorescent dyes. All dyes were numbered, and their corresponding chemical structures and properties were shown in Scheme S2 and Table S1. **Characters such as FD1080 and BTC1070 were adopted from their names in previous literatures.**

Reviewer #3 (Remarks to the Author):

General comment: Yan et al explored a new set of fluorophores which when compared to earlier ones are smaller and this brings a number of benefits. As the unique approach in their molecular design they make use of ground state antiaromaticity/excited state aromaticity. Yet, it is the computational investigations of the aromaticity effects that are the weakest in the manuscript; far below the quality one can expect to find in a paper published in a high-impact journal.

Response: We sincerely appreciate reviewer #3 for providing his/her insightful comments and suggestions that have improved the manuscript. Unfortunately, reviewer #3 questioned critically on the computational section, particularly on the employed computational method or theoretical indexes used to describe the (anti)aromaticity of studied molecules in this work. However, as we read these questions, we found most of the questions were caused by the misunderstandings on our presentation, which we have clarified in the revised manuscript.

Comment: 1) The aromaticity assessment is based on magnetic (NICS and ACID) and geometric aromaticity aspects (changes in BLA), yet this analysis is very unconvincing. First, it is written that NICS and ACID were calculated for the S1 state, however, these computations were apparently done with (what the authors call) the DeltaSCF method. No reference is given to this method, but I suspect that it is the approach which was used by Kawashima et al in <https://onlinelibrary.wiley.com/doi/10.1002/anie.201904882>, and heavily criticized by Karadakov and Saito in <https://onlinelibrary.wiley.com/doi/full/10.1002/anie.202001934>. That computational approach is easy but should never be used as the state which results is not a pure singlet state but a mix of singlet and triplet multiplicity. Much more useful would be a small CASSCF. Of course, the latter will not provide the authors with an ACID plot, but they can calculate NICS at CASSCF level with the Dalton program.

Response: First, the so-called Δ SCF method is indeed used in this work for the excited-state calculation and the associated properties as the reviewer mentioned. Herein Δ SCF refers to calculations using the difference between the ground-state Kohn-Sham (KS) energy and the energy of a KS determinant representing an excited configuration, with the energy being minimized self-

consistently with respect to the orbitals in each case [Ziegler, et al. *J. Theor. Chim. Acta* 1977, 43, 261; Hait and Head-Gordon, *J. Phys. Chem. Lett.* 2021, 12, 4517]. The notation Δ is used to indicate KS energy differences with (frozen) ground-state orbitals used in both KS determinants [Moore et al., *J. Chem. Theory Comput.* 2015, 11, 3305; Sotoyama, *J. Phys. Chem. A* 2021, 125, 48, 10373]. For example, considering a two-level model with an excitation involving a single occupied (*i*)-virtual (*a*) orbital pair (i.e., HOMO-LUMO), the linearized TDDFT singlet excitation energies are given as:

$${}^1\Delta E^{\text{TDDFT}} = (\varepsilon_a - \varepsilon_i) + [ai|f_{XC}^{\alpha\alpha} + f_{XC}^{\alpha\beta}|ai] + 2[ai|r_{12}^{-1}|ai]$$

where $f_{XC}^{\alpha\beta}(1,2)$ is the XC response kernel, ε_a and ε_i are ground-state orbital energies, “1” and “2” indicate the spatial coordinates of a pair of electrons. Similarly, the expressions for the singlet energies obtained from Δ SCF using density change $\Delta\rho$ for the excitation can be expressed as:

$${}^1\Delta E^{\text{SCF}} = (\varepsilon_a - \varepsilon_i) + \frac{1}{2}[\Delta\rho|r_{12}^{-1} + f_{XC}^{\alpha\alpha}|\Delta\rho] - [ii|f_{XC}^{\alpha\alpha} - f_{XC}^{\alpha\beta}|aa]$$

We greatly appreciate the reviewer for pointing this out, regarding the Δ SCF method in this work versus the anti-aufbau method as described in [Kawashima et al. *Angew. Chem. Int. Ed.* 2019, 58, 11686]. We believe that the two ones are quite similar but not exactly the same. The main difference comes from the employed DFT XC kernel. In the work by Kawashima et al., the typical B3LYP functional was employed, but in our work the optimally-tuned range-separated (OTRS) density functional LC-BLYP* whose range-separation parameter (ω) was optimally tuned according to Koopmans’ theorem. Our previous studies have demonstrated that such a OTRS method, more superior to those traditional hybrid functionals (i.e. PBE0, B3LYP et al.), can provide a balanced description between the electron delocalization and electron localization effects, and further produce reliable predictions for the electronic structures and other optoelectronic properties of organic molecular systems [see Sun et al. *J. Chem. Theory Comput.* 2015, 11, 3851; *J. Chem. Theory Comput.* 2015, 11, 3305; *J. Chem. Theory Comput.* 2014, 10, 1035; *J. Chem. Theory Comput.* 2016, 12, 2906; *J. Phys. Chem. Lett.* 2017, 8, 2393; *J. Comput. Chem.* 2016, 37, 684; *J. Comput. Chem.* 2017, 38, 569]. In addition, Kronik et al. demonstrated that the OTRS functional can offer a significant improvement in predicting accurate nuclear magnetic shielding (σ) values for a series of phosphine molecules over popular semi-local and hybrid density functionals [Kronik et al. *Adv. Theory Simul.*, 2020, 3, 2000083]. It should be noted that the performance of DFT methods in predicting nuclear magnetic resonance (NMR) related properties such as nucleus independent chemical shift (NICS) strongly depends on the specific molecular systems [Patra et al. *ChemPhysChem* 2021, 22, 298].

Further, it has been demonstrated for a long time that DFT-based Δ SCF with approximate DFT functionals (XC kernel) can consistently give improved performances in cases where conventional linear-response TDDFT fails. [see Kondo et al., *Nat. Photonics* 2019, 13, 678; Hait et al., *J. Chem.Theory Comput.* 2016, 12, 3353; Moore et al., *J. Chem. Theory Comput.* 2013, 9, 4991; Rudolph et al., *J. Chem. Phys.* 2011, 391, 92; Kowalczyk et al., *J. Chem. Phys.* 2011,134, 054128].

For the sake of safety, as the reviewer commented, the anti-aufbau method with B3LYP functional was used in [Kawashima et al. *Angew. Chem. Int. Ed.* 2019, 58, 11686] and criticized by [Karadakov and Saito, *Angew. Chem. Int. Ed.* 2020, 59, 9228] because it should not be used as the state is not a pure singlet state but a mix of singlet and triplet multiplicity. Per reviewer's suggestion, the calculations at the CASSCF(2,2)/6-31G(d) level were also performed using Dalton 2022 software as shown in the table below. It can be seen that all three methods give qualitatively consistent prediction results for the NICS(1)zz values for both S_0 and S_1 states, although the CASSCF-predicted value (-66.04) of S_1 is larger than those of DFT (-25.30 for LC-BLYP* and -23.87 for B3LYP). However, it should be noted that the CASSCF method may suffer from the issues such as choosing sufficient active space and electrons and lack of dynamic electron correlation [Ottosson et al. *ChemPlusChem* 2019, 84, 712]. We have included this calculation result in the revised manuscript (**shown below**).

In short, we greatly thank the reviewer for raising this method issue. The combined primary experimental evidences and supporting theoretical calculations consistently confirm the main conclusion of ground-state antiaromaticity and corresponding excited-state aromaticity for designed molecules in this work. And the main conclusion of the present study does not seem to be affected by the different methods. Additionally, we are confident that the Δ SCF method accompanying with the OTRS functional can improve the prediction of NMR properties (i.e. NICS) to some extent, compared to the common Δ SCF-B3LYP method. Obviously, this subtopic is worth further research which we may reserve as a subsequent study.

Page 16 in the main text:

Besides, to exclude the method dependencies, three methods including DFT-B3LYP, DFT-LC-BLYP* and complete-active-space self-consistent field (CASSCF) method were used for the selected AF3 dye (Table S3). Note that the CASSCF calculation is commonly considered as a standard approach for studying the magnetic properties of S_1 excited state [Ottosson et al. *ChemPlusChem* 2019, 84, 712; Karadakov et al. *Angew. Chem. Int. Ed.* 2020, 59, 9228-9230]. It can be seen that all

the three methods give qualitatively consistent predictions for the NICS(1)_{zz} values for both S₀ and S₁ states of AF3, supporting the main conclusion of this work.

Table S3 in the Supplementary Information:

Table S3. Calculated NICS(1)_{zz} values (ppm) based on the central five-membered ring of AF3 using three methods including B3LYP, LC-BLYP* and CASSCF(2,2) with a 6-31G(d) basis set. Note that NICS(1)_{zz} was defined as the zz component of the NICS value located 1 angstrom perpendicular to the plane above the five-membered ring.

State	B3LYP	LC-BLYP*	CASSCF(2,2)
S ₀	34.42	37.33	32.07
S ₁	-23.87	-25.30	-66.04

Comment: 2) With regard to the ACID plots in Figure S7 they are of so exceptionally low resolution that even if I magnify to 500% I cannot view the small current density arrows generated by the AICD program and which represent the alleged ring currents. The present plots are totally meaningless and indicate that the authors are unfamiliar with what is required for a proper (anti)aromaticity assessment.

Response: We apologize for the inconvenience caused by this ACID plots in Figure S7, in which its low resolution may arise from the pdf file compression during submission. Alternatively, we present high-resolution ACID plot of all AF dyes (Figure S7-S20, see Supplementary Information) by adding significant red arrows to show the current density.

Revised ACID plot of AF3 (Figure S9) for example:

Figure S9. Calculated AICD diagram of AF3 in S_0 and S_1 state.

Comment: 3) Coming to the NICS data, it has been clarified numerous times that single NICS values at specific points in space should never be used for polycyclic molecules. Instead one should use NICS-XY scans as clearly described by Gershoni-Poranne and Stanger in <https://chemistry-europe.onlinelibrary.wiley.com/doi/full/10.1002/chem.201304307> The present NICS data should be recalculated with that method.

Response: We thank the reviewer for this valuable advice. Per reviewer's suggestion, the NICS-XY scans as described in [Gershoni-Poranne et al., *Chem. Eur. J.*, 2014, 20, 5673] are carried out as shown below (**Figure R4**). The red line in the middle image shows the scanning path with a scanning interval of 0.1 Å. Positive values indicate anti-aromaticity, while negative values indicate aromaticity. The left image shows the NICS-XY scan at a height of 1 Å used in our work, i.e., NICS (1), while the right side is the 1.7 Å used in the reference, i.e., NICS (1.7). It can be seen that the overall trend is consistent, supporting the main conclusions in this work. We have also conducted NICS-XY scans (NICS(1)_{zz}) for all AF dyes (shown in Figure S21).

Figure R4. NICS-XY scans of AF3 in both S₀ (black) and S₁ (red) state. All the scans are performed at a height of 1 Å (left) and 1.7 Å (right). The red line in the middle image shows the scanning path with a scanning interval of 0.1 Å.

Figure S21 in the Supplementary Information:

Figure S21. NICS-XY scans of AF dyes in both S_0 (black) and S_1 (red) state. All the scans are performed at a height of 1 Å. The red line in the bottom right image shows the scanning path with a scanning interval of 0.1 Å.

Comment: 4) The analysis of the BLA values is odd; on page 5 of the manuscript is written that the

BLA is reduced to 1.374 – 1.463 Å, however, it does not tell from what range. Also, the CC bond length range 1.374 – 1.463 Å is quite large and not indicative of extensive Baird-aromaticity in the S1 state. Indeed, one can see in Figures 2e and 2f that the reduction in BLA upon excitation from S0 to S1 is rather modest.

Response: We thank the reviewer for this valuable comment and apologize for the misunderstandings caused by our presentation in the initial version of manuscript. In our revised manuscript, we applied the concept of bond length redistribution instead of BLA to describe the changes in bond lengths upon excitation, as BLA is primarily applicable to alternant hydrocarbons, not very suitable for non-alternant π -conjugated system such as the fluorene skeleton. This modification in the description did not alter our final conclusion.

Page 6 in the main text, the following paragraph was revised:

The C3-C4 bond that connects the two benzene rings in the fluorene skeleton is distinctly elongated at 1.486 Å, surpassing the lengths of other C-C bonds in the fluorene structure. These observations were confirmed in the optimized S₀ geometry (Figure 2e, 2f) based on the density functional theory (DFT) calculations. While for the optimized S₁ structure, nine C-C bonds undergo substantial bond length changes, centralizing on one side of the fluorene skeleton (C13-C1, C1-C2, C2-C3, C3-C4, C4-C5, C5-C6, C6-C7, C9-C4, C3-C11), with each bond experiencing a change in length between the S₀ and S₁ states exceeding 0.025 Å. We hypothesized that this bond length redistribution is linked to the changes in aromaticity, as for all AF dyes. Further, both nucleus-independent chemical shift scans along the X and Y axes (NICS-XY scan) and calculations of anisotropy of the induced current density (AICD) indicate a highly localized antiaromaticity in the five-member ring of the S₀ state, coupled with increased aromaticity in the entire fluorene skeleton of the S₁ state (Figures S7-S21). These results suggest the presence of an aromaticity reversal-induced vibronic coupling in the S₁-S₀ nonradiative transition.

Comment: 5) *Far too often do aromaticity analyses by experimentalists primarily rely on magnetic aromaticity indicators, however, there is a growing body of studies that show the difference between magnetic indices, on the one side, and electronic and energetic indices on the other (see e.g. <https://chemistry-europe.onlinelibrary.wiley.com/doi/full/10.1002/slct.201602080>). Thus, the*

authors should confirm the switch in aromatic and antiaromatic character upon excitation by some electronic index (e.g. MCI) which can be applied to both S0 and S1 (the latter at TD-DFT level).

Response: We thank the reviewer for the valuable comments regarding the various magnetic indices that may improve the theoretical analysis in this work. Per reviewer's suggestion, a series of electronic indexes used for characterizing magnetic aromaticity such as multicenter index (MCI), electron localization function-pi (ELF-pi) and Shannon aromaticity (SA) are calculated **as shown in Figure R5 below** using the Multiwfn code [Lu et al. *J. Comput. Chem.*, 2012, 33, 580]. First, for MCI [Giambiagi et al. *J. Phys. Chem. A*, 2006, 110, 7642; Pedersen et al. *RSC Adv.*, 2022, 12, 2830], the larger the MCI value, the stronger the aromaticity. It can be seen that the MCI values are in the order of S₁-TDDFT (0.105) > S₁- Δ SCF (0.103) > S₀ (0.102). However, it should be noted that the MCI cannot be used to determine the antiaromaticity since the antiaromatic systems typically possessing very small MCI that is indistinguishable from those with weak aromaticity [see Lu et al. *J. Comput. Chem.*, 2012, 33, 580 and its manual]. For ELF-pi [Santos et al. *JCP*, 2004, 120, 1670], the space enclosed by the higher ELF values makes it easier for electrons to delocalize within this space and simultaneously more difficult to populate outside of this space. The position indicated by the red arrow is called a bifurcation point, on which larger ELF-pi values suggest the more aromatic. According to the definition of ELF-pi value in [Poater et al. *Chem. Rev.* 2005, 105, 3911], the molecules with ELF-pi of 0.17~0.35 correspond to an antiaromatic character. Thus, it can be seen that the ELF-pi value of 0.18 for the S₀ state of AF3 indicates its antiaromaticity and the one of 0.43 for the S₁ state suggests a subtle aromatic character. For Shannon aromaticity (SA) [Noorizadeh et al. *PCCP*, 2010, 12, 4742], the SA index relies on the electronic information (Shannon) entropy at the bond critical points (BCP). The smaller the SA value the stronger the aromaticity, for example, the benzene molecule possesses the SA value of zero, indicating its good aromaticity. Typically, the SA value of < 0.003 corresponds to an aromatic character and the one of > 0.005 as antiaromaticity. The calculated SA value of 0.00192 for the S₁ state of AF3 confirms its significant aromaticity of S₁ state but the smaller value of 0.00183 for the S₀ state fails to predict its antiaromaticity of S₀ state.

To conclude, although the calculated electronic indexes for analyzing (anti)aromaticity can provide additional valuable information from the perspective of electron delocalization/population, it seems that the well-established calculations based on the nucleus-independent chemical shift (NICS) and anisotropy of the induced current density (AICD) can indeed produce direct and reliable

predictions of antiaromaticity of the S_0 state and aromaticity of the S_1 state for AF3 in this work.

Figure R5. Calculated electronic indexes used for characterizing magnetic aromaticity including multicenter index (MCI), electron localization function- π (ELF- π) and Shannon aromaticity (SA) whose detailed definition of formula can be found in Lu et al. *J. Comput. Chem.*, 2012, 33, 580 and the manual of Multiwfn code.

Comment: 6) The references on the excited state aromaticity are not properly selected and placed. The Chem Rev by Rosenberg et al on excited state (anti)aromaticity (ref 20) is used as a reference for Hückel-aromaticity in the ground state, and the purely computational study by Karadakov (ref 31) is used to support that it “has been experimentally proved that [Baird’s rule] likewise applies to the lowest singlet excited state”. It is obvious that the authors have not read the literature on excited state aromaticity properly. For experimental studies which support that Baird’s rule can be applied to the lowest singlet excited states I propose to cite <https://pubs.acs.org/doi/10.1021/ja00060a063> (together with <https://chemistry-europe.onlinelibrary.wiley.com/doi/full/10.1002/cplu.201900066>), <https://www.nature.com/articles/s41467-017-00382-1> and <https://pubs.acs.org/doi/full/10.1021/jacs.0c05611>. In addition to the Chem Rev by Rosenberg I propose to add the recent review by Yan et al (<https://chemistry-europe.onlinelibrary.wiley.com/doi/full/10.1002/chem.202203748>). The computational paper by Karadakov (ref 31) should definitely be given, yet I would also add <https://pubs.rsc.org/en/content/articlelanding/2011/cp/c1cp22239b> and

<https://pubs.acs.org/doi/10.1021/acs.joc.6b02460>

I also think they should cite Peter Wan's pioneering experimental work on photosolvolysis to fluorenyl cations, see <https://pubs.rsc.org/en/content/articlelanding/1985/c3/c39850001207>

Response: We greatly thank the reviewer for the careful reading and suggestions for the related references cited in this work. Per reviewer's suggestion, we accordingly cite these papers in the revised references (*shown below*).

Page 2 in the main text:

In the ground state, Hückel's rule rationalizes the enhanced stability of aromatic systems [Schleyer. *Chem. Rev.* 2001, 101, 1115-1118], and thus decrease HOMO energy.

Page 3 in the main text:

... and it has been computationally [Karadakov et al. *J. Phys. Chem. A* 2008, 112, 7303-7309; Feixas et al. *Phys. Chem. Chem. Phys.* 2011, 13, 20690-20703; Karadakov et al. *J. Org. Chem.* 2016, 81, 11346-11352] and experimentally [Shukla et al. *J. Am. Chem. Soc.* 1993, 115, 2990-2991; Ottosson et al. *ChemPlusChem* 2019, 84, 712-721; Ueda et al. *Nat. Commun.* 2017, 8, 346; Kotani et al. *J. Am. Chem. Soc.* 2020, 142, 14985-14992] proved that this rule likewise applies to the lowest singlet state. One may expect a naturally reduced HOMO-LUMO gap in a $[4n]\pi$ -electron system because ground-state antiaromaticity energetically raises HOMO level while excited-state aromaticity decreases LUMO level [Rosenberg et al. *Chem. Rev.* 2014, 114, 5379-5425; Rosenberg et al. *Chem. Eur. J.* 2014, 29, e202203748] (Scheme 1a right).

At the outset, we devised a compact chromophore unit featuring two amine groups linked to the C3 and C6 position of a cationic fluorene skeleton, which incorporates a 4π -electron five-membered ring as an antiaromatic core [Wan et al. *J. Chem. Soc., Chem. Commun.*, 1985, 1207-1208] (Scheme S1a).

Comment: 7) Finally, I wonder why they have used different functionals for different properties? The orbitals and orbital energies in Figure S3 were calculated with LC-BLYP* while those in Scheme 1 by B3LYP.

Response: Per reviewer's suggestion, the orbitals and orbital energies in the Scheme S1 are

recalculated and replotted with the consistent LC-BLYP* functional.

Scheme S1 in Supplementary Information:

Scheme S1. (b) HOMO/LUMO energy level and ΔE comparison of AF and pyronin skeleton shows the gap narrowing effect of antiaromatic skeleton introduction. The calculation was conducted using (TD)DFT at the LC-BLYP*/def-TZVP with DCM in PCM solvent model.

Comment: Summarized, the quality of the (excited state) aromaticity analysis is far below what can be expected from a manuscript submitted to a high-impact journal such as Nat. Commun. The aromaticity analysis is very speculative and insufficiently supported. There are many more items to address also, yet my time is simply limited. Those items will need to be addressed in further potential revisions.

Response: We are always very grateful to the reviewer for the efforts, professional suggestions and insightful comments that undoubtedly have improved our manuscript. We try our best to remove the misunderstandings on our presentation and address the issues as reviewer pointed out. We hope that we have clarified in the revised manuscript through the point-to-point response as shown above.

REVIEWER COMMENTS

Reviewer #1 (Remarks to the Author):

The manuscript has been improved and is ready for publication.

Reviewer #3 (Remarks to the Author):

The authors have added NICS results at CASSCF level, which was one of the items I requested in my earlier report, their ACID plots are of much better resolutions, and they have carried out NICS-XY scans. I am satisfied with these changes. What needs some more attention are their new results of the electronic aromaticity indices and their subsequent analysis. I hope what I write below is of help in strengthening this part of their study.

When I initially viewed the data from the electronic indices, I did a different evaluation than what the authors do. However, when considering a few things, I will possibly come to another conclusion after I see some further data. Their present results are inconclusive and they need to do a few additional computations, which all are straightforward.

First the MCI results: In their response letter, they report MCI values of the S0 and S1 states which are essentially the same (0.1025 for S0 with KS-DFT and 0.1051 for S1 with TDDFT). However, the dilemma is that they compare MCI results computed with Kohn-Sham DFT with those computed with TD-DFT, i.e., they compare "apples and oranges". When taking into account that TDDFT underestimates MCI values when compared to KS-DFT (see Table 3 in <https://chemistry-europe.onlinelibrary.wiley.com/doi/epdf/10.1002/cplu.201900066>) one can speculate that the MCI value of their S1 state should be higher than that of the S0 state, and then it may indicate some S1 aromatic character. The authors should do a few additional items: (i) they should instead compare MCI values computed at CASSCF(2,2) level for the S0 and S1 states, and (ii) they can check how similar the electron configuration of the T1 state is to that of the S1 state (apart from the multiplicity difference), and if the states are similar, they should report and analyze the MCI value obtained with KS-DFT for the T1 state and compare with the S0 state.

Then the ELFpi results: For the S1 state the authors report an ELFpi bifurcation value of 0.43, but in the review in Chem Rev from 2005, Poater et al writes "The values of ELFpi are in the 0.11-0.35 range for antiaromatic compounds, whereas they are >0.70 for aromatic compounds." Thus, the value 0.43 is well below anything that represents aromaticity (indeed closer to antiaromatic than aromatic). The dilemma is that the authors have selected the bifurcation value at one of the C-C bonds of the flanking benzene rings and not the ring-closure bifurcation value in the central 5-MR (which is what I would choose). Viewing the S1 state ELFpi surface in their response letter, it looks as if the ring-closure bifurcation value will be higher for the 5-MR, possibly within the range of aromaticity (i.e. above 0.70). Furthermore, if the

T1 and S1 states are similar with respect to electron configurations, they may want to look at the ELFpi of the T1 state which can be computed with KS-DFT similar as the S0 state. Indeed, usage of ELFpi to analyze the triplet state Baird-aromaticity of the cyclopentadienyl cation has been reported (see <https://chemistry-europe.onlinelibrary.wiley.com/doi/full/10.1002/cphc.200700540>), and they may want to compare against that as it would represent maximal Baird-aromaticity of a 5-MR.

At this point I would like to raise a final concern: The magnetic indices NICS and current densities from AICD suggest a reversal in the antiaromatic and aromatic upon excitation from S0 to S1, but the results from the electronic MCI and ELFpi indices do not. This may change with the additional data that I request, however, such divergent results have been observed gradually more in recent years (see list of papers below). Often magnetic aromaticity descriptors tell that a certain ring is aromatic or antiaromatic while electronic and/or energetic descriptors tell that the ring is nonaromatic. If the electronic indices still indicate the 5-MR in the compounds of the authors is nonaromatic in S1, then I recommend the authors to be open with this difference in their manuscript. This is much better than to tweak the conclusion towards a statement that there is an aromaticity/antiaromaticity reversal. Their paper will actually be of much higher scientific value if they are open with the difference in the results between the magnetic and electronic indices. In this context they may want to read section 2 and consider Figure 2 in the paper <https://chemistry-europe.onlinelibrary.wiley.com/doi/epdf/10.1002/chem.202203748>

Papers where differences between magnetic vs. electronic and energetic indices are observed:

<https://chemistry-europe.onlinelibrary.wiley.com/doi/full/10.1002/slct.201602080>

<https://onlinelibrary.wiley.com/doi/full/10.1002/poc.4455>

<https://pubs.acs.org/doi/full/10.1021/acs.joc.3c01807>

<https://pubs.acs.org/doi/full/10.1021/jacs.3c07335>

Finally, the results with electronic indices should be included into the ESI and mentioned in the manuscript. As the antiaromaticity/aromaticity reversal is central to the manuscript it is also odd that none of the figures in the manuscript contains any aromaticity information (apart from panels e and f in Figure 2).

Reviewer #4 (Remarks to the Author):

In this manuscript, the authors designed a new series of ultra-photostable aminofluorene (AF) dyes with mini structures for NIR-II imaging. Employing an antiaromatic design strategy, these dyes demonstrated advantages in various bioapplication scenarios. The manuscript exhibits logical coherence and meticulous organization, sufficient data support to substantiate the views elucidated by the authors. While the AF dyes may not be the brightest in comparison to existing NIR-II probes, this study is the first to highlight the potential of bioapplications based on an antiaromatic fluorophore and the methodology is sound. In the previous revision, the authors adequately addressed questions from other reviewers with more details provided, thereby enhancing the credibility of the data and improving the manuscript's quality. I therefore recommend acceptance of the manuscript after addressing several minor concerns

below.

1.The authors concentrated their studies on AF3 and its derivatives in the manuscript. Therefore, I suggest incorporating additional details about the photophysical properties of AF3, including absorption and emission spectra, quantum yield (QY), etc., in various solutions, particularly in PBS, which is biologically relevant. The data presented in Table 1 may not be sufficient to adequately highlight AF3.

2.The authors mentioned that AF3-COOH and AF3-Dextran could be made water-soluble through simple modification. It would be valuable to provide information on the water-solubility of the primitive AF3 and other AF dyes. Specifically, how does their water-solubility compare to existing NIR-II dyes like IR1048? This is noteworthy, especially considering the statement made by the authors regarding "larger dye skeletons likely resulting in poor solubility." Please provide additional insights on this aspect.

3.In Figure S6, the authors compared the photostability of AF3 and IR1048 under 200 kW cm⁻² laser irradiation. While this high-power density is effective for highlighting the photostability of AF3, it may not be representative of suitable application scenarios, especially in cellular environments where the photothermal effect of such high-power density could lead to cell death. It is recommended that the authors comment on the application potential of the high-power density appropriately in their manuscript.

4.On page 7, line 183, the authors mentioned, "though, the absorbance of AF4, AF7 and AF11 showed a decline when the GSH concentration exceeded 100 μM," while in Figure S22, it appears that AF13 rather than AF11 exhibited a decline. Please clarify this discrepancy in the text for accuracy.

5.In Figure 3b, the plot appears to be somewhat confusing. I suggest using either a dot-line or a bar graph for both Signal-to-Background Ratio (SBR) and intensity to enhance clarity. Additionally, the representation of black lines and yellow bars in Figure 3d and 3e is unclear. It would be helpful to clearly indicate whether they represent the mean ± standard deviation (s.d.) in the figure caption for better interpretation.

Reviewer #1 (Remarks to the Author):

General comment: The manuscript has been improved and is ready for publication.

Response: We sincerely appreciate the reviewer #1 for the positive comment.

Reviewer #3 (Remarks to the Author):

General comment: The authors have added NICS results at CASSCF level, which was one of the items I requested in my earlier report, their ACID plots are of much better resolutions, and they have carried out NICS-XY scans. I am satisfied with these changes. What needs some more attention are their new results of the electronic aromaticity indices and their subsequent analysis. I hope what I write below is of help in strengthening this part of their study.

When I initially viewed the data from the electronic indices, I did a different evaluation than what the authors do. However, when considering a few things, I will possibly come to another conclusion after I see some further data. Their present results are inconclusive and they need to do a few additional computations, which all are straightforward.

Response: We greatly appreciate reviewer #3 for the positive comments on our revisions and also providing his/her insightful suggestions. Per reviewer's suggestions, we have made point-to-point responses to the questions raised by reviewer and hope that these issues left have been clarified in the revised manuscript.

Comment: 1) First the MCI results: In their response letter, they report MCI values of the S0 and S1 states which are essentially the same (0.1025 for S0 with KS-DFT and 0.1051 for S1 with TDDFT). However, the dilemma is that they compare MCI results computed with Kohn-Sham DFT with those computed with TD-DFT, i.e., they compare "apples and oranges". When taking into account that TDDFT underestimates MCI values when compared to KS-DFT (see Table 3 in <https://chemistry-europe.onlinelibrary.wiley.com/doi/epdf/10.1002/cplu.201900066>) one can speculate that the MCI value of their S1 state should be higher than that of the S0 state, and then it may indicate some S1 aromatic character. The authors should do a few additional items: (i) they should instead compare MCI values computed at CASSCF(2,2) level for the S0 and S1 states, and (ii) they can check how

similar the electron configuration of the T1 state is to that of the S1 state (apart from the multiplicity difference), and if the states are similar, they should report and analyze the MCI value obtained with KS-DFT for the T1 state and compare with the S0 state.

Response: We thank the reviewer for this valuable advice and totally agree with the additional calculations for MCI index. Per reviewer's suggestion, firstly the MCI values for both S₀ and S₁ states are computed at the CASSCF(2,2)/6-31G(d) level. It can be seen that the MCI values from CASSCF calculations are too small due to the insufficient dynamic electron correlation [*ChemPlusChem* 2019, 84, 712–721]. The MCI value of the S₁ state is indeed higher than that of the S₀ state, indicating some S₁ aromatic character.

Table S4. MCI indexes (in electrons) for AF3 in the S₀, S₁, and T₁ states computed at various levels.

Methods	S ₀	S ₁	T ₁
LC-BLYP*	0.1025	0.1030 (Δ SCF)	0.1126 (UDFT) ^b
		0.1051/0.1039 (TD/TDA) ^a	0.1092 (TDA) ^a
B3LYP	0.0654	0.0670 (TD)	0.0726 (UDFT) ^b
CASSCF(2,2)	0.0159	0.0183	0.0224

^athe Tamm–Dancoff approximation (TDA) scheme of TD-DFT is shown to reliably predict triplet excited state versus traditional TD-DFT. ^bunrestricted density functional theory (UDFT).

Furthermore, considering that the typical TDDFT underestimates MCI values versus KS-DFT, the excited-state characters for both S₁ and T₁ are revealed through the analysis of the natural transition orbitals (NTOs) and the electron configurations (see Figure S21a below). The results suggest that the character of T₁ state is quite similar to that of S₁ state, and the NTOs of T₁ state by TDA-DFT looks almost the same as its singly occupied molecular orbitals (SOMOs). Thus, the MCI values are also calculated for both S₁ and T₁ states (Table S4). The UDFT-calculated MCI values of T₁ state are obviously larger than that of S₀ state, suggesting more aromatic character of excited state.

Figure S21. (a) The analysis of the natural transition orbitals (NTOs) using TDA-DFT and the electron configurations based on singly occupied molecular orbitals (SOMOs) using UDFT.

In the Supporting Information, we add the Table S4, Figure S21 and the following resulting comments as supplementary note for Figure S21:

For MCI [Giambiagi et al. *J. Phys. Chem. A*, 2006, 110, 7642; Pedersen et al. *RSC Adv.*, 2022, 12, 2830], the larger the MCI value, the stronger the aromaticity. Considering the fact that the typical TDDFT underestimates MCI values for S₁ state, the MCI values for T₁ state possessing similar excited state characters vs. S₁ state (Figure S21a) are obtained using unrestricted KS-DFT method. It can be seen that the MCI values are in the order of T₁-UDFT (0.1126) > S₁-TD/TDA (0.1051/0.1039) > S₁- Δ SCF (0.1030) > S₀ (0.1025). In addition, the MCI values for both S₀ and S₁ states are also computed at the CASSCF(2,2)/6-31G(d) level. All the calculated MCI values of T₁ states are larger than that of S₀ state, suggesting more aromatic character of excited state. Last but not least, it should be noted that the MCI is not a good choice to determine the antiaromaticity since the antiaromatic systems typically possessing very small MCI that is indistinguishable from those with weak aromaticity [see Lu et al. *J. Comput. Chem.*, 2012, 33, 580 and its manual].

Figure S21. (b) Calculated electronic indexes based on multicenter index (MCI) and electron localization function- π (ELF- π) used for characterizing aromaticity in the five-membered ring of AF3 and detailed definition of formula can be found in Lu et al. *J. Comput. Chem.*, 2012, 33, 580 and the manual of Multiwfn code.

Comment: 2) Then the ELFpi results: For the S_1 state the authors report an ELFpi bifurcation value of 0.43, but in the review in *Chem Rev* from 2005, Poater et al writes "The values of ELFpi are in the 0.11-0.35 range for antiaromatic compounds, whereas they are >0.70 for aromatic compounds." Thus, the value 0.43 is well below anything that represents aromaticity (indeed closer to antiaromatic than aromatic). The dilemma is that the authors have selected the bifurcation value at one of the C-C bonds of the flanking benzene rings and not the ring-closure bifurcation value in the central 5-MR (which is what I would choose). Viewing the S_1 state ELFpi surface in their response letter, it looks as if the ring-closure bifurcation value will be higher for the 5-MR, possibly within the range of aromaticity (i.e. above 0.70). Furthermore, if the T_1 and S_1 states are similar with respect to electron configurations, they may want to look at the ELFpi of the T_1 state which can be computed with KS-DFT similar as the S_0 state. Indeed, usage of ELFpi to analyze the triplet state Baird-aromaticity of the cyclopentadienyl cation has been reported (see <https://chemistry-europe.onlinelibrary.wiley.com/doi/full/10.1002/cphc.200700540>), and they may want to compare

against that as it would represent maximal Baird-aromaticity of a 5-MR.

Response: Per reviewer's suggestion, the electronic index such as electron localization function- π (ELF- π) [Santos et al. *JCP*, 2004, 120, 1670] used for characterizing aromaticity are calculated based on the five-member ring of AF3 (Figure S21b). The space enclosed by the higher ELF values makes it easier for electrons to delocalize within this space and simultaneously more difficult to populate outside of this space. The position indicated by the red arrow is called a bifurcation point, on which larger ELF- π values suggest the more aromatic. According to the definition of ELF- π value in [Poater et al. *Chem. Rev.* 2005, 105, 3911], the molecules with ELF- π of 0.17~0.35 correspond to an antiaromatic character, and ELF- π of > 0.70 to aromatic character. Thus, it can be seen that the ELF- π value of 0.18 for the S_0 state of AF3 indicates its antiaromaticity character. And the ELF- π value for the S_1 state is calculated to be 0.47 using the Δ SCF method. Note that usage of ELF- π to analyze the triplet state Baird-aromaticity is proved by Ottosson et al. [*ChemPhysChem*, 2008, 9, 257], herein the ELF- π value for T_1 state is calculated to be 0.37 using the unrestricted KS-DFT method. Unfortunately, these obtained ELF- π values for excited states are significantly smaller than the reference value of 0.70, suggesting not aromatic characters at all.

In the Supporting Information, we add the Figure S21 and the following resulting comments as supplementary note for Figure S21:

For ELF- π [Santos et al. *JCP*, 2004, 120, 1670] index, the space enclosed by the higher ELF values makes it easier for electrons to delocalize within this space and simultaneously more difficult to populate outside of this space. The position indicated by the red arrow is called a bifurcation point, on which larger ELF- π values suggest the more aromatic. According to the definition of ELF- π value in [Poater et al. *Chem. Rev.* 2005, 105, 3911], the molecules with ELF- π of 0.17~0.35 correspond to an antiaromatic character, and ELF- π of > 0.70 to aromatic character. Thus, it can be seen that the ELF- π value of 0.18 for the S_0 state of AF3 indicates its antiaromaticity character (Figure S21b). And the ELF- π value for the S_1 state is calculated to be 0.47 using the Δ SCF method. Note that usage of ELF- π to analyze the triplet state Baird-aromaticity is proved by Ottosson et al. [*ChemPhysChem*, 2008, 9, 257], herein the ELF- π value for T_1 state is calculated to be 0.37 using the unrestricted KS-DFT method. Unfortunately, these obtained ELF- π values for excited states are significantly smaller than the reference value of 0.70, suggesting not aromatic characters at all.

Comment: 3) At this point I would like to raise a final concern: The magnetic indices NICS and current densities from AICD suggest a reversal in the antiaromatic and aromatic upon excitation from S_0 to S_1 , but the results from the electronic MCI and ELFpi indices do not. This may change with the additional data that I request, however; such divergent results have been observed gradually more in recent years (see list of papers below). Often magnetic aromaticity descriptors tell that a certain ring is aromatic or antiaromatic while electronic and/or energetic descriptors tell that the ring is nonaromatic. If the electronic indices still indicate the 5-MR in the compounds of the authors is nonaromatic in S_1 , then I recommend the authors to be open with this difference in their manuscript. This is much better than to tweak the conclusion towards a statement that there is an aromaticity/antiaromaticity reversal. Their paper will actually be of much higher scientific value if they are open with the difference in the results between the magnetic and electronic indices. In this context they may want to read section 2 and consider Figure 2 in the paper <https://chemistry-europe.onlinelibrary.wiley.com/doi/epdf/10.1002/chem.202203748>

Papers where differences between magnetic vs. electronic and energetic indices are observed:

<https://chemistry-europe.onlinelibrary.wiley.com/doi/full/10.1002/slct.201602080>

<https://onlinelibrary.wiley.com/doi/full/10.1002/poc.4455>

<https://pubs.acs.org/doi/full/10.1021/acs.joc.3c01807>

<https://pubs.acs.org/doi/full/10.1021/jacs.3c07335>.

Response: We thank the reviewer for this advice and providing closely-related references. Based on the additional calculations for electronic indexes, it can be seen that the magnetic indices NICS and current densities from AICD suggest a reversal in the antiaromatic and aromatic upon excitation from S_0 to S_1 , while the results from the electronic MCI and ELF- π indexes do not. Per reviewer's suggestion, we leave the difference between magnetic and electronic indexes as an open question for this article. For the consistency and difference between the magnetic and electronic indices used for characterizing (anti-)aromaticity, we refer the interested readers to, e.g., refs [H. Ottosson et al, *Chem. Eur. J.* 2023, 29, e202203748; *J. Am. Chem. Soc.* 2023, 145, 41, 22527; G. Frenking et al, *ChemistrySelect* 2017, 2, 863]

Page 6 in the main text, the following paragraph has been added:

Moreover, a series of electronic indexes used for characterizing (anti-)aromaticity such as

multicenter index (MCI) and electron localization function- π (ELF- π) are calculated (Table S4, Figure S21). It can be seen that the magnetic indices NICS and current densities from AICD indeed suggest a reversal in the antiaromatic and aromatic upon excitation from S_0 to S_1 , while the results from these electronic indexes (i.e. MCI and ELF- π) do not (see supporting information for more details). For the consistency and difference between the magnetic and electronic indices used for characterizing (anti-)aromaticity, we refer the interested readers to, e.g., refs [H. Ottosson et al, *Chem. Eur. J.* 2023, 29, e202203748; *J. Am. Chem. Soc.* 2023, 145, 41, 22527; G. Frenking et al, *ChemistrySelect* 2017, 2, 863].

Comment: 4) Finally, the results with electronic indices should be included into the ESI and mentioned in the manuscript. As the antiaromaticity/aromaticity reversal is central to the manuscript it is also odd that none of the figures in the manuscript contains any aromaticity information (apart from panels e and f in Figure 2).

Response: Per reviewer's suggestion, the above results regarding the electronics indices such as MCI and ELF- π are included in the electronic supporting information. The AICD and NICS-XY scan results of AF5 are also included in revised Figure 2g-2i (shown below).

Revised Figure 2g-2i:

Figure 2 | Photophysical mechanistic investigation of AF3. ... g-h, theoretical calculated AICD diagram of AF3 in S_0 (g) and S_1 (h) state. i, NICS(1) $_{zz}$ -XY scan result of AF3 in both S_0 and S_1 state. The red line in the inset shows the scanning path with a scanning interval of 0.1 Å. ...

Reviewer #4 (Remarks to the Author):

General comment: In this manuscript, the authors designed a new series of ultra-photostable aminofluorene (AF) dyes with mini structures for NIR-II imaging. Employing an antiaromatic design strategy, these dyes demonstrated advantages in various bioapplication scenarios. The manuscript exhibits logical coherence and meticulous organization, sufficient data support to substantiate the views elucidated by the authors. While the AF dyes may not be the brightest in comparison to existing NIR-II probes, this study is the first to highlight the potential of bioapplications based on an antiaromatic fluorophore and the methodology is sound. In the previous revision, the authors adequately addressed questions from other reviewers with more details provided, thereby enhancing the credibility of the data and improving the manuscript's quality. I therefore recommend acceptance of the manuscript after addressing several minor concerns below.

Response: We sincerely appreciate the reviewer #4 for the positive comments and valuable suggestions. In response, we conducted a comprehensive revision, addressed the concerns raised by reviewer (shown below). We believe this revision has significantly improved the manuscript.

Comment: 1) The authors concentrated their studies on AF3 and its derivatives in the manuscript. Therefore, I suggest incorporating additional details about the photophysical properties of AF3, including absorption and emission spectra, quantum yield (QY), etc., in various solutions, particularly in PBS, which is biologically relevant. The data presented in Table 1 may not be sufficient to adequately highlight AF3.

Response: We appreciate the reviewer for the comments. In this revision, we have added a photophysical property table of AF3 in different solvents (**Table S2** in revised Supplementary Information, shown below).

Table S2. Photophysical properties of AF3 in different solvents.

Solvent	λ_{abs} (nm)	ϵ ($\text{M}^{-1} \text{cm}^{-1}$)	λ_{em} (nm)	Stokes shift (nm)	Quantum yield (10^{-2} %)
DCM	959	23052	1014	55	1.1
Toluene	946	27288	993	47	2.0
Dioxane	940	23765	997	57	1.4
DMF	949	18773	1015	66	0.9

DMSO	951	19618	1022	71	1.1
EtOH	950	20368	1006	55	0.7
PBS	942	20764	1022	80	0.4

Comment: 2) The authors mentioned that AF3-COOH and AF3-Dextran could be made water-soluble through simple modification. It would be valuable to provide information on the water-solubility of the primitive AF3 and other AF dyes. Specifically, how does their water-solubility compare to existing NIR-II dyes like IR1048? This is noteworthy, especially considering the statement made by the authors regarding "larger dye skeletons likely resulting in poor solubility." Please provide additional insights on this aspect.

Response: We appreciate the reviewer for the comments. As shown in Figure S30, AF1, AF2 and AF3 showed non-distorted PA signal in a wide range of dye concentrations (50-200 μM), which indicates that for small-molecule AF dyes, their water solubility reaches at least 200 μM . Although we observed a decrease of water-solubility for other AF dyes with larger N-substituents (e.g., AF14), all AF dyes showed normal absorption spectra with no distortion in PBS solution compared with those in DCM (Figure S2) in a concentration of 20 μM , while IR1048 is not water-soluble even in a low concentration of 1 μM (only soluble in 1%DMSO/1 \times PBS, see "Cell permeability comparison by live-cell imaging and quantification" in Methods part). In summary, primitive AF3 and other AF dyes are also water-soluble, significantly better than IR1048. We have modified the following paragraph to emphasize the water-solubility of AF dyes in our revised manuscript:

Page 12 in the main text, the following paragraph has been modified:

First, we tested the photoacoustic (PA) readouts of several AF dyes (AF1, AF2 and AF3 were tested due to the limit of instrumental excitation) with a water-soluble PA benchmark indocyanine (ICG, commercially available). In a parallel comparison, AF dyes exhibited stable PA spectra shape correlating with molecular absorption and showed linearly increasing PA signals in a wide range of dye concentrations (50-200 μM), while ICG displayed distorted PA spectra shape and nonlinear signals (Figure S30). This result suggests that the superior solubility of AF dyes in water confers stable spectral behavior, which are suitable for in vivo MSOT.

Comment: 3) In Figure S6, the authors compared the photostability of AF3 and IR1048 under 200 kW cm⁻² laser irradiation. While this high-power density is effective for highlighting the photostability of AF3, it may not be representative of suitable application scenarios, especially in cellular environments where the photothermal effect of such high-power density could lead to cell death. It is recommended that the authors comment on the application potential of the high-power density appropriately in their manuscript.

Response: We appreciate the reviewer for the comments. Regarding the applicability of this high-power density in cellular environment, we did not find any deformation of fixed cells due to photothermal effect under continuous 200 kW cm⁻² laser irradiation (see Supplementary Video 1), indicating that such a higher-power density can be used in a fixed cell environment. We acknowledge that, due to the lower brightness, NIR-II AF dyes are not competitive with commercial live cell dyes for fluorescent imaging. However, we believe that this high-power density durability has greater application potential in some single-molecule research areas. As written in our manuscript, “it is worthwhile to explore additional single-molecule photophysical properties of AF dyes beyond fluorescence, such as Raman scattering, absorption, and photothermal and photoacoustic effects”, this high-power density has greater application potential in many single-molecule research fields. In single-molecule localization microscopy, high power density (>50 kW cm⁻²) is required to maintain good SNR and localization precision [Nat. Rev. Methods Primers 2021, 1, 39], and in Raman spectroscopy analysis, the laser power density could reach a high level up to 10⁵-10⁶ W cm⁻² [Chem. Rev. 2017, 117, 11, 7583], these facts collectively highlight the necessity of high-power density stable probes in many scenarios, particularly in single-molecule detection. In revised manuscript, we have modified following discussion on the application potential of the high-power density:

Page 14 in the main text, the following paragraph has been modified:

... This property may have potential applications in the field of single-molecule science that necessitates strong light-matter interaction, For example, single-molecule localization microscopy and Raman spectroscopy analysis always require high power density irradiation up to 10⁴-10⁶ W cm⁻² [Nat. Rev. Methods Primers 2021, 1, 39; Chem. Rev. 2017, 117, 11, 7583]. In this context, it is worthwhile to explore additional single-molecule photophysical properties of AF dyes beyond fluorescence, such as Raman scattering, absorption, and photothermal and photoacoustic effects.

Comment: 4) On page 7, line 183, the authors mentioned, "though, the absorbance of AF4, AF7 and AF11 showed a decline when the GSH concentration exceeded 100 μ M," while in Figure S22, it appears that AF13 rather than AF11 exhibited a decline. Please clarify this discrepancy in the text for accuracy.

Response: We appreciate the reviewer for identifying inaccuracies in our manuscript. In our revised manuscript, we have corrected this mistake.

Comment: 5) In Figure 3b, the plot appears to be somewhat confusing. I suggest using either a dot-line or a bar graph for both Signal-to-Background Ratio (SBR) and intensity to enhance clarity. Additionally, the representation of black lines and yellow bars in Figure 3d and 3e is unclear. It would be helpful to clearly indicate whether they represent the mean \pm standard deviation (s.d.) in the figure caption for better interpretation.

Response: We appreciate the reviewer for the comment. In our revised Figure 3, we have clearly indicated the parameters that each bar refers to in Figure 3b to avoid misunderstanding. The figure caption of Figure 3d and 3e were also modified according to reviewer's suggestion.

Revised Figure 3b:

Figure 3 | Cell permeability and photostability characterization. ... b, Quantified data of the signal-to-noise ratio (SBR) and mean fluorescence intensities for AF3-stained cells at different time points. The bars represent mean \pm s.d. derived from n = 5 independent cells. ...

Revised Figure caption of 3d and 3e:

Figure 3 | Cell permeability and photostability characterization. ... d, Repetitive fluorescence recording of AF3-stained endothelial cells at a laser power density of 200 kW cm^{-2} . Right panel shows the signal-to-noise ratio (SBR) of cell images versus irradiation time. Dips in SBR curve were mainly caused by the changes of instrument focal plane. The black lines and yellow bars represent mean \pm s.d. derived from $n = 5$ independent cells. **e,** Control result using Cy7-stained cells. The black lines and yellow bars represent mean \pm s.d. derived from $n = 5$ independent cells.

REVIEWER COMMENTS

Reviewer #3 (Remarks to the Author):

I still have two remarks which the authors must address before I think the manuscript can be accepted. First, when viewing the NICS-XY scan in Figure 2i I am puzzled why there is a maximum in the center of the 5-membered ring. An aromatic species should have a minimum in the NICS-XY scan at that position. However, when viewing the AICD plots in the Supporting Information one can see local circulations (vortices) at the transannular C-C bonds which likely counteracts the diatropic (aromatic) ring currents in the perimeter. I propose the authors to add in the manuscript the sentence "Local circulations in the current density at the transannular bonds seem to slightly weaken the aromatic character in the 5-membered ring." It would further be good if they can tell in the caption of Figure 2 that higher resolution AICD plots are found in the Supporting Information.

Secondly, in the Abstract I would like the authors to tell that it is magnetic descriptors that suggest an aromaticity/antiaromaticity reversal. That it not clear in the present Abstract and a reader may believe that the aromaticity/antiaromaticity reversal has been shown unambiguously (which is not the case, the electronic indices tell something else). Thus, I would like them to include "as indicated by magnetic descriptors," in the sentence starting "We find that aromaticity reversal ..." so that this sentence reads "We find that aromaticity reversal upon electronic excitation, as indicated by magnetic descriptors, not only ...". In short, the authors have presently only weak support for the antiaromaticity/aromaticity reversal which they claim to observe (the ELF-pi results do not support their claim, which should have been the case). In a follow-up study I strongly advice the authors to report a more comprehensive analysis using further electronic indices, e.g. EDDB, and where they may have to take back their present claim.

Reviewer #4 (Remarks to the Author):

In this new version manuscript, the authors have carefully revised the manuscript and properly addressed the concerns raised by the reviewers. By right, the manuscript has been improved already, and the current manuscript can be thus considered for acceptance without further moderation.

Reviewer #3 (Remarks to the Author):

Comment: 1) I still have two remarks which the authors must address before I think the manuscript can be accepted. First, when viewing the NICS-XY scan in Figure 2i I am puzzled why there is a maximum in the center of the 5-membered ring. An aromatic species should have a minimum in the NICS-XY scan at that position. However, when viewing the AICD plots in the Supporting Information one can see local circulations (vortices) at the transannular C-C bonds which likely counteracts the diatropic (aromatic) ring currents in the perimeter. I propose the authors to add in the manuscript the sentence “Local circulations in the current density at the transannular bonds seem to slightly weaken the aromatic character in the 5-membered ring.” It would further be good if they can tell in the caption of Figure 2 that higher resolution AICD plots are found in the Supporting Information.

Response: We greatly appreciate the reviewer #3 for the suggestions. Per reviewer’s suggestion, we have modified corresponding paragraphs in our revised manuscript.

Page 6 in the main text, the following paragraph has been modified:

Further, both nucleus-independent chemical shift scans along the X and Y axes (NICS-XY scan) and calculations of anisotropy of the induced current density (AICD) indicate a highly localized antiaromaticity in the five-member ring of the S₀ state, coupled with increased aromaticity in the entire fluorene skeleton of the S₁ state, although local circulations in the current density at the transannular bonds seem to slightly weaken the aromatic character in the 5-membered ring.

The caption of Figure 2 has been modified:

Figure 2 | Photophysical mechanistic investigation of AF3. ... g-h, theoretical calculated AICD diagram of AF3 in S₀ (g) and S₁ (h) state. For high-resolution AICD plots of all AF dyes, please refer to Figures S7-S20 in the Supplementary Information. ...

Comment: 2) Secondly, in the Abstract I would like the authors to tell that it is magnetic descriptors that suggest an aromaticity/antiaromaticity reversal. That it not clear in the present Abstract and a reader may believe that the aromaticity/antiaromaticity reversal has been shown unambiguously (which is not the case, the electronic indices tell something else). Thus, I would like them to include “, as indicated by magnetic descriptors,” in the sentence starting “We find that aromaticity reversal ...”

so that this sentence reads “We find that aromaticity reversal upon electronic excitation, as indicated by magnetic descriptors, not only ...”. In short, the authors have presently only weak support for the antiaromaticity/aromaticity reversal which they claim to observe (the ELF- π results do not support their claim, which should have been the case). In a follow-up study I strongly advice the authors to report a more comprehensive analysis using further electronic indices, e.g. EDDB, and where they may have to take back their present claim.

Response: Per reviewer’s suggestion, we have modified related sentences in Abstract and Discussion part to accurately express our aromaticity calculation results.

Page 2 in the main text, the following sentence has been modified:

... **We find that aromaticity reversal upon electronic excitation, as indicated by magnetic descriptors,** not only reduces the energy bandgap but also induces strong vibronic coupling, resulting in ultrafast excited-state dynamics and unparalleled photostability.

Page 13 in the main text, the following sentence has been modified:

... **Our quantum chemical calculations on magnetic descriptors have revealed** the aromaticity reversal between the S_0 and S_1 states, which leads to a strong vibration coupling within the antiaromatic fluorene skeleton and a high rate of non-radiative internal conversion.

Reviewer #4 (Remarks to the Author):

General comment: In this new version manuscript, the authors have carefully revised the manuscript and properly addressed the concerns raised by the reviewers. By right, the manuscript has been improved already, and the current manuscript can be thus considered for acceptance without further moderation.

Response: We sincerely appreciate the reviewer #4 for the positive comment.

REVIEWERS' COMMENTS

Reviewer #3 (Remarks to the Author):

I am happy with the final revisions carried out by the authors and think the manuscript is now ready to be accepted for publication.